behaviour/ecology

foraging behaviour, behavioural consistency, GPS tracking, foraging ecology, Australasian gannet, accelerometry

**Author for correspondence:**
John P. Y. Arnould
e-mail: john.arnould@deakin.edu.au

# Geographical, temporal and individual factors influencing foraging behaviour and consistency in Australasian gannets

Marlenne A. Rodríguez-Malagón, Elodie C. M. Camprasse, Lauren P. Angel and John P. Y. Arnould

School of Life and Environmental Sciences (Burwood Campus), Deakin University, Geelong, Victoria 3220, Australia

Foraging is a behaviour that can be influenced by multiple factors and is highly plastic. Recent studies have shown consistency in individual foraging behaviour has serious ecological and evolutionary implications within species and populations. Such information is crucial to understand how species select habitats, and how such selection might allow them to adapt to the environmental changes they face. Five foraging metrics (maximum distance from the colony, bearing from the colony to the most distal point, tortuosity index, total number of dives and mean vectorial dynamic body acceleration were obtained using GPS tracking and accelerometry data in adult Australasian gannets (*Morus serrator*) from two colonies in southeastern Australia. Individuals were instrumented over two breeding seasons to obtain data to assess factors influencing foraging behaviour and behavioural consistency over multiple timescales (consecutive trips, breeding stages and years) and habitats (pelagic, mixed pelagic and inshore, and inshore). Colony, breeding stage and year were the factors which had the greatest influence on foraging behaviour, followed by sex. Behavioural consistency, measured as the contribution of the individual to the observed variance, was low to moderate for all foraging metrics (0.0–27.05%), with the higher values occurring over shorter timescales. In addition, behavioural consistency was driven by spatio-temporal factors rather than intrinsic characteristics. Behavioural consistency was higher in individuals foraging in inshore than pelagic habitats or mixed pelagic/inshore strategy, supporting suggestions that consistency is favoured in stable environments.

# 1. Introduction

Foraging is a primary activity of animals which can be highly influenced by intrinsic factors such as age, sex or genotype [1–3], extrinsic factors such as geographical location, local weather or predation risk [4–6], and by reproductive constraints such as breeding stage or brood size [7,8]. However, such factors do not necessarily affect all animals in the same way, with one or multiple influences potentially acting in different directions at a particular time [9]. Furthermore, as foraging is generally a high energy expenditure activity, there are strong incentives for animals to develop foraging strategies that minimize their energy costs [10]. Such strategies can vary in associated foraging time and effort in particular habitats, associated choice of specific search methods and/or of food types consumed [11]. If a particular foraging strategy provides greater rewards, it is likely that this strategy will be repeated over time, favouring the development of behavioural consistency [12]. It is known that behavioural consistency in foraging activities leads to the evolutionary development of foraging specialization within animal populations, but the information on the persistence of this phenomenon over different timescales and habitats is limited [13].

Foraging specialization refers to the use of a specific proportion of the full range of available resources (or foraging strategies) by a subset of a population, resulting in inter-individual niche variation [14]. This phenomenon has been demonstrated in a wide variety of taxa [15]. Information on the factors influencing behavioural consistency (e.g. extrinsic versus intrinsic factors) and on the link between habitat selection (e.g. pelagic versus benthic foraging) and behavioural consistency is lacking; however, foraging specializations are thought to arise in stable environments in which resources are predictable and diverse, enabling individuals to develop behavioural differences to reduce niche overlap with conspecifics and, thus, minimize competition [16]. Such behavioural consistency may, therefore, have significant ecological consequences at the individual level but also on the development of offspring during the breeding season [17]. Consequently, knowledge of foraging temporal and spatial variation in specializations is important to fully understand their ecological implications within species [18,19].

The marine environment is complex and dynamic, and the foraging ecology of marine life is highly influenced by environmental variables [20]. At a global scale, oceans display clear patterns of water circulation and climate [21]. At local scales, physical features such as bathymetry, tidal regimes and nutrient fluxes determine the structure of marine and coastal ecosystems and influence the behaviour of marine fauna [22]. Marine environments comprise different ecosystem types and biomass levels which can lead to the development of a wide range of foraging techniques (higher ecological opportunity) [16,23] even within the same species and populations [24]. Behavioural consistency in foraging activities have been found within different animal groups in the marine environment [25,26], and it is expected to occur more commonly for top-order marine predators who are regulated by bottom-up processes and experience high levels of resource competition [27,28].

Marine birds are important top-order predators [29,30]. They are long-lived animals and, during the breeding season, adopt a central place foraging strategy which can lead to high levels of resource competition [31]. These attributes have been shown to favour the development of behavioural consistency within this group and, combined with other factors such as age, sex or breeding status, influence the development of individual behavioural differences [32]. However, the degree to which species and populations exhibit individual behavioural consistency in foraging activities can vary [33]. Studies suggest that intra- and inter-population differences may be related to temporal changes in resource availability [32,34,35], but the mechanisms influencing individual foraging consistency across populations or habitats are poorly understood [36]. Such information is crucial to enable predictions about marine predators' habitat selection and the responses of natural populations to changing environments [18,37].

The Australasian gannet (*Morus serrator*) is an important marine predator in southeastern Australia and New Zealand [38,39], with an estimated annual consumption of 228.2 tons of schooling pelagic fish (e.g. Australian sardine *Sardinops sagax*, barracouta *Thyrsites atun* and blue mackerel *Scomber australasicus*) in Australian waters alone [40]. This region is one of the fastest warming oceanic areas and significant changes to ocean currents are predicted to occur [41,42]. Such changes are likely to alter the distribution and abundance of marine species [43,44]. Indeed, expansions and shifts in fish and invertebrate species ranges have already been documented in southeastern Australia [45]. Therefore, knowledge of the factors influencing foraging activity and behavioural consistency in Australasian gannets is necessary to understand how their populations may adapt to changes in the supply of marine resources.

Like other members of the Sulidae family, the Australasian gannet is considered a generalist forager and has been shown to be adaptable in its feeding habits [40,46,47]. It displays reverse sexual

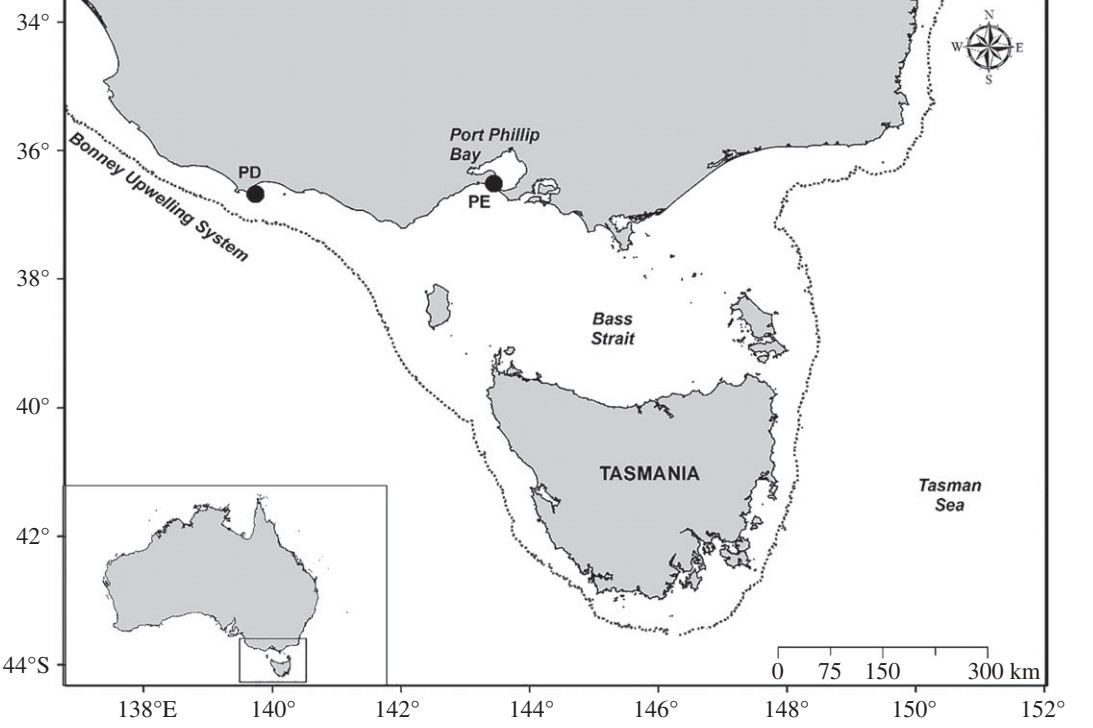

**Figure 1.** Location of study sites: Point Danger (left) and Pope's Eye (right). The 200 m bathymetric contour is given to indicate the edge of the continental shelf.

dimorphism (females larger than males) and recent studies suggest individuals exhibit sex-related differences in habitat use [48–50] and prey selection [47]. Furthermore, inter-colony differences in diving behaviour and habitat selection have been documented [51,52]. However, little is known of the factors influencing foraging behaviour and behavioural consistency in this species. The aims of the present study, therefore, were to determine, in Australasian gannets: (i) the influence of extrinsic and intrinsic factors on foraging behaviour, (ii) the degree of behavioural consistency in foraging activities, (iii) the persistence of behavioural consistency through time and space, and (iv) the influence of extrinsic and intrinsic factors on behavioural consistency.

# 2. Material and methods

## 2.1. Study sites and animal handling

The study was conducted at two Australasian gannet breeding colonies in northern Bass Strait, southeastern Australia, which experience divergent oceanographic conditions and may present differences in resource availability or habitat accessibility (figure 1). Point Danger (PD; 38°23′36.09″ S, 141°38′55.94″ E) is located at the western edge of Bass Strait near the seasonally active (Austral summer) Bonney Upwelling, an important source of primary productivity for the Bass Strait region [22,53]. Individuals from this colony range up to 238 km northwest and southeast, remaining over the narrow (approx. 40 km wide) continental shelf to forage on schooling fish and cephalopods [22,52]. Pope's Eye (PE; 38°16′35.88″ S, 144°41′56.21″ E) is located at the entrance of the Port Phillip Bay (PPB) on an artificial structure. Birds from PE forage within the shallow (average depth less than 13.6 m) [54] waters of PPB primarily on benthic/demersal fish, outside of PPB within northern Bass Strait on schooling fish and cephalopods, or in both habitats [49,52].

Data were collected during the 2014/15 and 2015/16 breeding seasons (October–March) in each of three breeding stages: incubation; early chick-rearing (chick age 0–50 days); and late chick-rearing (chick age greater than 50 days) [55]. Individuals were captured by hand or with the aid of a noose-pole [56] at the nest and weighed in a cloth bag with a suspension scale (±25 g, Salter Australia Pty Ltd, Australia). A GPS data logger (programmed to record location every 2 min; I-gotU GT-600, Mobile Action Technologies Inc., Taiwan, ±10 m error), and a tri-axis accelerometer data logger

(sampling rate of the individual 25 Hz; X8M-3mini, Gulf Coast Data Concepts LLC, USA), encapsulated together in heat shrink plastic (total package 53.7 g, <3% body mass), were then attached to the central tail feathers of the individual using water-proof tape (Tesa® 4651, Beiersdorf AG, Germany). The instrumentation was done in a consistent manner, with the GPS logger at the top, towards the head of the animals, and the accelerometer at the bottom towards the tail. Individuals were then returned to the nest and resumed natural behaviours within 10 min of capture.

After 10–12 days, individuals were recaptured as previously described and the data loggers were removed by peeling the tape from the feathers, and body mass was recorded. Morphometric measurements of culmen length and bill depth, and tarsus length and ulna length, were taken using Vernier callipers (± 0.1 mm) and metal ruler (± 1 mm), respectively. A blood sample (0.1 ml) was then obtained by venipuncture of a tarsal vein for genetic sexing (DNA Solutions, Wantirna South, Victoria, Australia) before the bird was returned to the nest. Where possible, the same individuals were sampled in multiple breeding stages and across years.

## 2.2. Data processing and statistical analysis

Unless stated otherwise, all data processing and statistical analyses were conducted in R v. 3.3.2 [57]. Deployment data were checked by visual inspection, and split into individual foraging trips using the return of the birds to the colonies' coordinates as endpoints for each trip. Trips were then filtered using a speed filter to remove erroneous locations [58] in the *trip* package [59], applying a maximum average speed of 55 km h$^{-1}$ suggested for northern gannets (*M. bassanus*) [34]. Subsequently, for each foraging trip, maximum distance from the colony (km), total distance travelled (km) and bearing (0–360°, from the colony to the most distal point) were calculated using the *adehabitatHR* package [60]. A tortuosity index, a measure of an animal's searching behaviour, was also estimated by dividing the maximum distance reached from the colony with the total distance travelled during the trip [61,62].

At-sea behaviours throughout the foraging trip were determined from the tri-axis accelerometer data loggers. Data were initially inspected visually to assign foraging behaviours (plunge diving and surface foraging) in IGOR Pro (v. 6.37, WaveMetrics, USA) [63], based on the acceleration profiles suggested for other species of gannets and boobies [64–66]. The *Ethographer* package was then used to identify these behaviours by performing a *k*-means, unsupervised cluster analyses of 1 s windows of continuous wavelet spectra computed from the time series. Later, each identified cluster was assigned a specific behaviour based on the previous visual identification [67]. From these data, the total number of dives (plunge diving and surface foraging) was estimated for each foraging trip. In addition, the accelerometry data were used to calculate the average vectorial dynamic body acceleration (VeDBA) throughout each foraging trip. This average was used as a proxy of the energy expenditure and allowed comparisons of the rate of energy expended across foraging trips [68–70].

A body condition index (BCI) was calculated for each bird at each deployment, as a proxy for total body fat (%) content, using body mass (kg), wing ulna (mm) and tarsus (mm) measurements [48]. As Australasian gannets are sexually dimorphic, body size indices were calculated to investigate the effect of size on foraging behaviour independently of sex. A body size index (BSI) and wing length index (WLI) were calculated using the deviation of each individual's body mass (kg) and wing length (mm) from the means for their respective sex.

To determine the factors influencing the foraging behaviour of instrumented individuals, linear mixed effects models were created using the *nlme* package [71]. The foraging metrics (maximum distance from the colony, average bearing and tortuosity index), the total number of dives and mean VeDBA, were used separately as response variables. Fixed factors such as colony (PD, PE), year (2014/15, 2015/16), breeding stage (incubation, early chick-rearing, late chick-rearing) and sex were used as explanatory variables in combination with the BCI, BSI and WLI. For these models, the full dataset was used (i.e. data obtained from all individuals in both sites and 2 years of sampling) using sampling size of three or more foraging trips per deployment. In this study, the influence of the factors colony, breeding stages and year were considered to reflect the differences in resource availability and environmental variation, respectively. Consequently, specific environmental variables were not analysed.

Where appropriate, variables were cube-root-transformed to fit model assumptions of constant variance and normal distribution of residuals [72]. Model assumptions were checked by plotting residuals and using quantile–quantile plots. Collinearity among all the explanatory variables was checked before conducting each model using pairplots, boxplots and the variance inflation factor (cut-off value used = 2) [73]. The initial models were then fitted with restricted maximum likelihood (REML) and models with and without the random structure (nest identity, due to the use of breeding partners

from the same nests, and individual identity) were compared using the *anova* function. Variance structure for the explanatory variables was included when the residuals inspection suggested it was necessary. The best-fixed structure was found using the *dredge* function of the *MuMIn* package based on the AICc values [74], using models refitted with maximum likelihood (ML). Where multiple models had ΔAICc ≤ 4 and no single model had an AICc weight above 0.90, model averaging was used to calculate the relative importance of each explanatory variable using the *MuMIn* package [74,75]. This multi-model statistical approach was selected as it allows to identify strong associations between multiple explanatory variables, while AICc values compare multiple models all at once incorporating model selection uncertainty and enabling inferences that are unconditional on a specific model [76,77].

To quantify the magnitude of individual behavioural consistency in each foraging metric, variance component analyses were conducted using the models containing the parameters defined as influential after model averaging. The *ape* package [77] was used to calculate the variance, standard deviation and proportion of total variance occurring at the individual level, as well as the residual variation. The variance explained by the individual is considered an estimate of the individual specialization within a population [15,19].

To investigate the factors influencing individual variation, a second set of models using another measure of consistency, the coefficients of variation of the foraging metrics (standard deviation in the case of the bearing as it is a circular variable), calculated per deployment and used as response variable, was then developed [78]. The same set of explanatory variables and modelling approach described above were used.

As multiple logger deployments were performed on most individuals (mean ± s.e.: 1.9 ± 0.1 deployments per bird), the full dataset allowed comparisons at different timescales to be made: trip-to-trip (T-to-T, data from consecutive trips obtained within the same deployment), breeding stage-to-breeding stage (S-to-S, data obtained from different breeding stages within the same year), and year-to-year (Y-to-Y, data obtained from the same breeding stage in two different years). This partition enabled the assessment of the timescales over which individual behavioural consistency is maintained. The full dataset was then partitioned to match each timescale tested, using in each case three or more foraging trips per deployment.

For the T-to-T comparison, the dataset was also subdivided into colonies (PD and PE) to quantify the individual consistency level at each site. The PD data were analysed in its entirety, reflecting the relatively uniform foraging habitat used by these individuals, whereas the PE data were split according to the predominant habitat individuals foraged in. Individuals that spent greater than 70% of trips during a deployment in PPB or Bass Strait were classified as PE-inshore and PE-pelagic, respectively, while individuals that did not were defined as PE-mixed. This partition allowed a comparison of individual behavioural consistency associated with different habitat selection. Models from these further separations were made using the same set of response and explanatory variables and followed all considerations described previously. Unless otherwise stated, results are presented as mean ± s.e.

# 3. Results

## 3.1. Factors influencing foraging behaviour

The GPS and accelerometry data loggers were deployed on 142 breeding birds (260 deployments) from which 3–50 foraging trips were obtained (18.1 ± 0.9 trips per individual). From the GPS data loggers, a total of 2493 foraging trips were recorded but, due to battery life restrictions, accelerometry data were recorded for only 1284 trips. Consequently, the sample size used to obtain the different foraging metrics varied depending on the device used. A summary of the calculated foraging metrics is presented in table 1.

The top-ranked statistical models explaining the factors influencing the foraging behaviour of Australasian gannets were determined using model averaging, as the combined weight of the top set of models was low ($\omega i < 0.9$, electronic supplementary material, tables S1 and S2). After model averaging, the best explanatory variables for maximum distances from the colony were colony, breeding stage, year and sex. Namely, individuals from PD, individuals during incubation, individuals during 2015/16 and females reached greater distances from the colony. For bearing and tortuosity index, colony, breeding stage and year were the most influential variables. Specifically, individuals from PD, individuals during late chick-rearing and individuals during 2015/16 showed higher bearings and tortuosity indices. Lastly, mean VeDBA and number of dives per foraging trip were both

**6**

**Table 1.** Means ± s.e. of the foraging trip parameters collected from instrumented breeding Australasian gannets (*Morus serrator*) during two years 2014/15 and 2015/16 at the Point Danger (PD) and Pope's Eye (PE) colonies in Victoria, Australia. Instrumentation in birds lasted 10–12 consecutive days. Data represent the dataset acquired summarized by sex, breeding stages (incubation = INC, early chick-rearing = ECR, late chick-rearing = LCR), year and colony-habitat. Samples sizes of the metrics estimated from the GPS (maximum distances from the colony, bearing and tortuosity index; $n_1$) and accelerometer data loggers (mean vectorial dynamic body acceleration and number of dives, $n_2$) are shown.

| | | $n_1$ | distances from colony (km) | bearing (°) | tortuosity index | $n_2$ | mean VeDBA (g) | Number of dives |
|---|---|---|---|---|---|---|---|---|
| sex | ♂ | 1424 | 60.41 ± 1.76 | 192.5 ± 2.0 | 0.29 ± 0.01 | 666 | 0.63 ± 0.21 | 345.7 ± 11.8 |
| | ♀ | 1056 | 77.22 ± 1.98 | 201.4 ± 2.0 | 0.30 ± 0.01 | 618 | 0.57 ± 0.17 | 390.6 ± 12.8 |
| breeding stage | INC | 452 | 95.47 ± 4.08 | 186.2 ± 4.1 | 0.26 ± 0.01 | 205 | 0.54 ± 0.16 | 584.0 ± 28.4 |
| | ECR | 1143 | 65.58 ± 1.78 | 193.9 ± 2.0 | 0.30 ± 0.01 | 619 | 0.61 ± 0.19 | 313.6 ± 9.5 |
| | LCR | 885 | 55.88 ± 1.90 | 204.6 ± 2.3 | 0.30 ± 0.01 | 460 | 0.61 ± 0.20 | 342.8 ± 14.6 |
| year | 2014 | 1224 | 64.18 ± 2.09 | 187.8 ± 2.2 | 0.27 ± 0.01 | 390 | 0.62 ± 0.21 | 398.1 ± 19.4 |
| | 2015 | 1256 | 70.86 ± 1.64 | 204.8 ± 1.9 | 0.31 ± 0.01 | 894 | 0.59 ± 0.19 | 353.7 ± 9.2 |
| colony-habitat | PD–pelagic | 1162 | 93.31 ± 2.32 | 234.1 ± 1.9 | 0.32 ± 0.01 | 686 | 0.59 ± 0.18 | 369.8 ± 12.5 |
| | PE–pelagic | 699 | 56.77 ± 1.61 | 181.0 ± 1.6 | 0.29 ± 0.01 | 375 | 0.58 ± 0.17 | 369.1 ± 14.8 |
| | PE–mixed | 355 | 39.46 ± 1.75 | 150.8 ± 4.1 | 0.26 ± 0.01 | 145 | 0.62 ± 0.22 | 401.4 ± 26.2 |
| | PE–inshore | 264 | 20.63 ± 1.68 | 131.1 ± 3.6 | 0.22 ± 0.01 | 78 | 0.74 ± 0.29 | 273.4 ± 29.6 |

influenced the most by breeding stage and sex. Individuals during early chick-rearing and males had higher VeDBA values, and individuals during incubation and females displayed a higher number of dives (table 2).

## 3.2. Influence of timescales and habitats on foraging behaviour consistency

Variance component analyses were performed to determine the proportion of variance explained by the individual for each of five foraging metrics. As 86% ($n = 226$) of the deployments were conducted simultaneously on breeding partners from the same nests, nest identity was tested during the modelling as a random component. The addition of this random component did not significantly improve models in all cases ($p > 0.05$ in all cases) and was, therefore, unnecessary. Conversely, the individual random component was significant ($p < 0.05$) in all but two of the sets of models developed. For short-term comparisons (T-to-T comparisons), the variance associated with the individual component ranged from low to moderate (11.1–27.1%) and overall decreased as the timescale comparison increased to mid-term (S-to-S: 9.5 to 22.9%) and long-term (Y-to-Y: 0.0 to 28.6%, table 3).

For PE, 127 of the deployments were categorized according to the predominant habitat in which each bird foraged, with 70 classified as PE-pelagic, 33 as PE-mixed and 24 as PE-inshore (see examples in figure 2). The proportion of females for each classification was 69%, 27% and 4%, respectively, with males being more abundant at PE-inshore and PE-mixed. The proportion of variance explained by the individual between habitats overall ranged from low to moderate values (3.2–50.4%). Consistency values (maximum distances from the colony, tortuosity indices, number of dives) were higher for both pelagic habitats (PD-pelagic, PE-pelagic) compared to the mixed strategy (PE-mixed). Except for one variable (VeDBA), consistency values were higher for the inshore strategy (PE-inshore) (table 4).

Using the coefficients of variation (or the standard deviations) of foraging metrics within deployments as a measure of individual consistency, the factors influencing individual variation were investigated. The five foraging metrics examined (maximum distance from the colony, bearing from the colony to the most distal point, tortuosity index, mean VeDBA and total number of dives) required model averaging due to the lack of a single best model from the candidate set of models. After model averaging, the most influential factors on individual variation identified were year, colony and breeding stage for the T-to-T comparison level, with individuals sampled during year 2014/15, at PD and in late chick-rearing stage having the highest consistency (electronic supplementary material, table S3).

## 4. Discussion

Determining the factors influencing foraging behaviour in marine predators and the persistence of behavioural consistency through time is crucial to understand habitat selection and how populations can adapt to fast environmental changes [78]. In the present study, spatio-temporal factors (colony, stage, year) influenced the foraging behaviour metrics obtained in breeding Australasian gannets the most, while individual characteristics (BCI, BSI and WLI) did not, with the exception of sex. The proportion of variation explained by the individual showed higher values over shorter (T-to-T) than longer (S-to-S and Y-to-Y) timescales, consistent with previous studies investigating the persistence of behavioural consistency in seabirds [78,79] and the repeatability of behaviours in several taxa [80]. Individual consistency in foraging behaviour was found to be higher in inshore compared to pelagic habitats and mixed use of both habitats, supporting suggestions that consistency is favoured in stable environments with predictable resources [79,81]. Lastly, measures of individual variation (CVs and SDs) were explained by spatio-temporal factors rather than individual characteristics; this supports the idea that consistency is linked to the strategies displayed by individuals depending on habitat selection and prey availability dictated by environmental variables rather than intrinsic factors.

### 4.1. Factors influencing foraging behaviour

The use of metrics in foraging ecology research to describe the behaviour and estimate energetic expenditure of animals, particularly marine birds and mammals, and to provide an indication of the foraging strategies and habitats used, is common practice [82]. Colony, year and breeding stage were the most influential factors on the foraging metrics analysed in the present study (i.e. maximum distance from the colony, bearing, tortuosity index, mean VeDBA and number of dives), followed by sex. Indices of body condition and body size (BCI, BSI, WLI) did not influence these metrics.

**Table 2.** Most parsimonious models after model averaging, and their corresponding estimated regression parameters, for five foraging metrics obtained from instrumented breeding Australasian gannets (*Morus serrator*).

| foraging metric | model fixed effects | fixed effect | estimate | s.e. | d.f. | t-value | p-value |
|---|---|---|---|---|---|---|---|
| distance from colony (km) | colony + stage + year + sex | (intercept) | 4.25 | 0.09 | 2340 | 47.58 | <0.0001 |
| | | colony (PE) | −0.79 | 0.09 | 134 | −8.65 | <0.0001 |
| | | stage (INC) | 0.51 | 0.06 | 2340 | 8.18 | <0.0001 |
| | | stage (LCR) | −0.21 | 0.06 | 2340 | −3.48 | 0.0005 |
| | | sex (male) | −0.32 | 0.09 | 134 | −3.54 | 0.0005 |
| | | year (2015) | 0.13 | 0.04 | 2340 | 3.08 | 0.002 |
| bearing (°) | colony + stage + year | (intercept) | 226.72 | 4.71 | 2353 | 48.15 | <0.0001 |
| | | colony (PE) | −71.88 | 6.22 | 135 | −11.55 | <0.0001 |
| | | stage (INC) | −16.91 | 3.98 | 2353 | −4.24 | <0.0001 |
| | | stage (LCR) | 11.73 | 3.81 | 2353 | 3.07 | 0.002 |
| | | year (2015) | 14.82 | 2.97 | 2353 | 4.97 | <0.0001 |
| tortuosity index | colony + stage + year | (intercept) | 0.31 | 0.005 | 2353 | 66.70 | <0.0001 |
| | | colony (PE) | −0.05 | 0.006 | 135 | −8.18 | <0.0001 |
| | | stage (INC) | −0.05 | 0.005 | 2353 | −10.07 | <0.0001 |
| | | stage (LCR) | 0.01 | 0.005 | 2353 | −0.55 | 0.58 |
| | | year (2015) | 0.04 | 0.004 | 2353 | 11.20 | <0.0001 |
| mean VeDBA | stage + sex | (intercept) | 0.83 | 0.01 | 1175 | 140.90 | <0.0001 |
| | | stage (INC) | −0.04 | 0.01 | 1175 | −5.55 | <0.0001 |
| | | stage (LCR) | −0.01 | 0.01 | 1175 | −0.07 | 0.95 |
| | | sex (male) | 0.03 | 0.01 | 108 | 3.70 | 0.0003 |
| number of dives | stage + sex | (intercept) | 6.62 | 0.14 | 1127 | 47.76 | <0.0001 |
| | | stage (INC) | 1.48 | 0.16 | 1127 | 9.12 | <0.0001 |
| | | stage (LCR) | 0.19 | 0.14 | 1127 | 1.40 | 0.16 |
| | | sex (male) | −0.36 | 0.17 | 106 | −2.04 | 0.04 |

**Table 3.** Variance component analysis of instrumented breeding Australasian gannets *(Morus serrator)*. Short- (trip-to-trip), medium- (stage-to-stage) and long-term (year-to-year) comparisons are shown. Sample sizes (number of trips/number of individuals) are presented for each final model. The significant fixed components of the models for which the coefficients of variation were used as response variable are shown as the factors influencing the individual variation in each case.

| foraging trip parameter | timescale | $\sigma^2$ | $\Sigma$ | $\sigma^2$ (%) | n | influences on individual variation |
|---|---|---|---|---|---|---|
| distances from colony (km)* | T-to-T | 0.21 | 0.46 | 27.05 | 2480/137 | colony, year |
| | S-to-S | 0.14 | 0.37 | 13.00 | 1166/56 | colony, breeding stage |
| | Y-to-Y | 0.166 | 0.40 | 16.64 | 1069/53 | sex |
| bearing (°) | T-to-T | 1090.48 | 33.02 | 26.32 | 2490/137 | breeding stage, year |
| | S-to-S | 944.32 | 30.72 | 22.90 | 1184/57 | breeding stage |
| | Y-to-Y | 1195.88 | 34.58 | 28.63 | 1069/53 | none |
| tortuosity index | T-to-T | 0.0006 | 0.026 | 11.07 | 2490/137 | colony, sex, year, WLI |
| | S-to-S | 0.0005 | 0.024 | 9.45 | 1184/57 | colony, BSI, sex |
| | Y-to-Y | 0.0005 | 0.023 | 8.70 | 1069/53 | WLI, year, sex |
| mean VeDBA (g)* | T-to-T | 0.001 | 0.005 | 16.91 | 1237/108 | stage, sex |
| | S-to-S | 0.001 | 0.005 | 14.66 | 641/51 | none |
| | Y-to-Y | — | — | 0.00 | 190/15 | NA |
| number of dives* | T-to-T | 0.57 | 0.75 | 18.89 | 1237/108 | colony, breeding stage, year |
| | S-to-S | 0.55 | 0.74 | 16.25 | 641/51 | none |
| | Y-to-Y | — | — | 0.00 | 190/15 | NA |

*Cube-root-transformed variables.

Geographical variation has previously been reported in the foraging behaviour of gannets [51,52,83,84] and other marine predators, reflecting spatial differences in resource availability or habitat accessibility [32,85,86]. The results of the present study are consistent with these findings and reveal the substantial differences in oceanographic regimes and habitats available to individuals from the PD and PE gannet colonies [49,52]. In particular, the individuals from PD, which forage within the Bonney Upwelling system, had longer foraging trips and higher tortuosity index than individuals sampled at PE, consistent with previous findings [52].

Year of sampling was found to influence the foraging behaviour of breeding Australasian gannets with individuals travelling less, having a lower tortuosity index, higher energy expenditure rate and diving more often during the 2014/15 compared to the 2015/16 breeding season. Breeding success (proportion of chicks fledged) was lower in 2014/15 (25% versus 50% at PE, and 48% versus 79% at PD, respectively; Rodríguez-Malagón 2014–2016, unpublished data). This suggests both sites experienced similar environmental variation influencing both foraging behaviour and reproductive success in a similar way. Previous studies at PE have reported an increased foraging effort in years of low local marine productivity [87], and inter-annual variation in foraging behaviour in response to environmental perturbations have been observed in other gannet species [83,88,89]. Indeed, primary productivity (as measured by chlorophyll-a concentration) was substantially higher in 2014/15 than in 2015/16 [90], coinciding with a strong El Niño-Southern Oscillation event with sea surface temperatures above average (bom.gov.au) in the later year.

Differences in foraging metrics were also evident between the different stages of the breeding season. Individuals conducted longer foraging trips, had a higher tortuosity index, lower energy consumption rate and dived more during incubation compared to the later breeding stages. Similar observations have been made in Australasian gannets [40,87] and other seabirds, and are thought to reflect a shift from self-feeding during incubation behaviour to chick-provisioning [32]. However, other studies have related changes in foraging behaviour between incubation and chick-rearing to be in response to temporal variation in prey availability due to environmental changes around colonies throughout the breeding period [91–93].

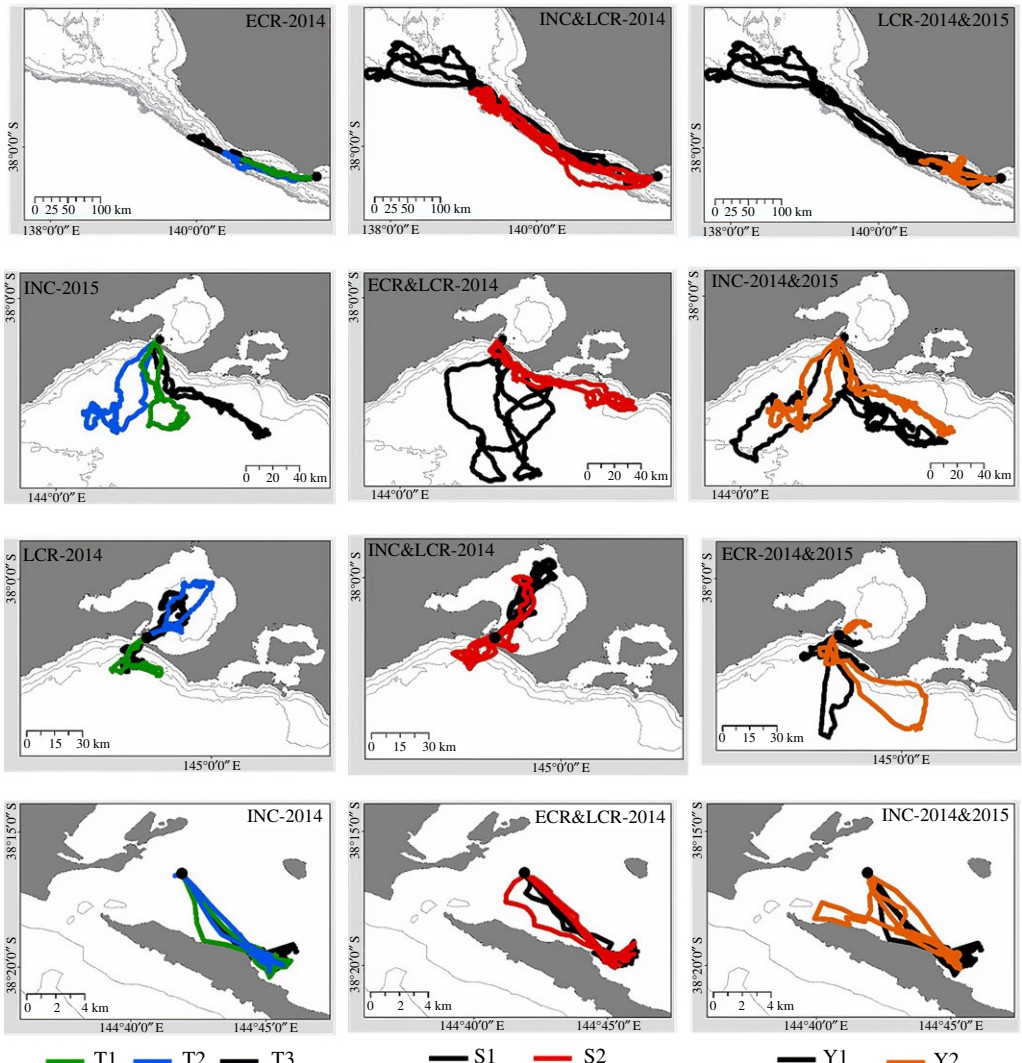

**Figure 2.** Examples of the time-scale comparisons investigated: T-to-T (trip-to-trip, left column); S-to-S (stage-to-stage, middle column); and Y-to-Y (year-to-year, right column). Each row represents an instrumented bird: first row, male from Point Danger-pelagic; second row, female from Pope's Eye-pelagic; third row, male from Pope's Eye-mixed; and fourth row, male from Pope's Eye-inshore. Breeding stages include incubation (INC), early chick-rearing (ECR) and late chick-rearing (LCR).

Maximum distance from the colony and the number of dives were shown to be influenced by sex, with females exhibiting higher values than males. Previous research at the two study colonies has shown sex differences in core foraging areas with only 4.2% and 18.4% of overlap at PD and PE, respectively [52]. Australasian gannets display reverse sexual dimorphism, with females being significantly heavier and larger than males [48]. This is consistent with observations in other Sulidae species in which males forage closer inshore than females, the larger sex [1,94,95]. In species with sexual size dimorphism, trophic or spatial segregation can function to reduce intra-specific competition, particularly during periods of intense resource competition [32,89]. Despite the greater foraging range and higher dive rate, females in the present study had lower mean VeDBA. This suggests females may be more efficient in some aspects of their foraging behaviour. Indeed, females from the study colonies have been previously been reported to spend a greater proportion of their foraging trips in gliding rather than flapping flight [52,87].

## 4.2. Influence of timescales and habitats on foraging behaviour consistency

The level of behavioural consistency displayed by individuals is thought to be related to the foraging strategy they adopt and influence how adaptable they can be when faced with rapid environmental changes [78,96]. Individuals in the present study displayed overall low to moderate levels of

**Table 4.** Variance component analysis of instrumented Australasian gannets (*Morus serrator*). Short time-scale comparison (trip-to-trip) results are shown of the models split by colony and habitat. Sample sizes (number of trips/number of individuals) are presented for each final model. The significant fixed components of the models for which the coefficients of variation were used as response variable are shown as the factors influencing the individual variation in each case.

| foraging trip parameter | | | | |
| --- | --- | --- | --- | --- |
| colony-habitat | $\sigma^2$ | $\Sigma$ | $\sigma^2$ (%) | *n* |
| distances from colony (km)* | | | | |
| PD-pelagic | 0.24 | 0.49 | 15.93 | 1170/76 |
| PE-pelagic | 0.019 | 0.13 | 3.17 | 704/41 |
| PE-mixed | 0.227 | 0.47 | 20.22 | 355/25 |
| PE-inshore | 0.28 | 0.53 | 50.37 | 264/16 |
| bearing (°) | | | | |
| PD-pelagic | 1171.45 | 34.22 | 26.32 | 1170/76 |
| PE-pelagic | 443.88 | 21.06 | 23.48 | 704/41 |
| PE-mixed | 1032.96 | 32.13 | 14.74 | 355/25 |
| PE-inshore | 2200.57 | 46.91 | 49.18 | 264/16 |
| tortuosity index | | | | |
| PD-pelagic | 0.0005 | 0.02 | 8.92 | 1170/76 |
| PE-pelagic | 0.0004 | 0.02 | 8.53 | 704/41 |
| PE-mixed | 0.0015 | 0.038 | 10.84 | 355/25 |
| PE-inshore | 0.0010 | 0.031 | 12.21 | 264/16 |
| mean VeDBA (g)* | | | | |
| PD-pelagic | 0.001 | 0.03 | 14.81 | 668/54 |
| PE-pelagic | 0.001 | 0.03 | 15.48 | 353/36 |
| PE-mixed | 0.001 | 0.02 | 9.73 | 149/18 |
| PE-inshore | 0.01 | 0.04 | 12.56 | 76/10 |
| number of dives* | | | | |
| PD-pelagic | 0.24 | 0.49 | 8.00 | 668/54 |
| PE-pelagic | 0.49 | 0.70 | 20.17 | 353/36 |
| PE-mixed | 0.98 | 0.99 | 26.03 | 149/18 |
| PE-inshore | 0.96 | 0.98 | 41.62 | 76/10 |

*Cube-root-transformed variables.

behavioural consistency in foraging metrics. As time between sampling increased, behavioural consistency decreased.

Over the short-term (T-to-T), breeding Australasian gannets in the present study displayed moderate levels of consistency, specifically in distances from the colony and bearing to most distal point, which suggest some degree of foraging side fidelity during consecutive trips, and potentially the exploitation of the same resource patches, similar to what has been found in northern gannets [97] and other seabirds [32,79,96]. By contrast, tortuosity index, mean VeDBA and number of dives were shown to be less consistent, suggesting that while individuals tended to revisit patches, they could adapt to current local prey availability and environmental conditions, rather than be limited by individual morphology or foraging/diving abilities.

Consistent with other studies in marine top-order predators [78,79], behavioural consistency in the present study decreased over time; it was higher for consecutive trips, compared to between breeding stages, and between years. Behavioural consistency in foraging implies individuals learn, remember and select specific resources and foraging strategies [98]. It requires predictability in the abundance and location of the exploited resources so that the strategies can be maintained in the population [36], which is less likely to be maintained over time because of environmental variability. Our results

support the theory that behavioural consistency can only persist as long as stability in environmental conditions prevails [99]. Importantly, as seabird foraging conditions are highly susceptible to fluctuations in the environment [20,100,101], it may be advantageous for individuals to maintain a certain level of behavioural plasticity to respond to such change [97]; behavioural consistency can lead marine predators to encounter ecological traps in degraded environments, and limit the adaptability of individuals to environmental changes [97,102,103].

In the present study, differences in consistency were seen when breeding Australasian gannets selected different foraging habitats. Birds which foraged in pelagic environments at both colonies (PD-pelagic and PE-pelagic) were less consistent than birds which used both pelagic and inshore environments (PE-mixed), which in turn were less consistent than foraging in inshore environments (PE-inshore). Individual consistency is thought to be promoted in stable environments [81,104,105]. Birds in the PE-inshore category foraged in PPB, a shallow environment with an important coverage of seagrasses and sandy bottoms that represent important habitats for marine invertebrates and fish in southeastern Australia [106]. Such benthic environments are indeed considered refuges for fish and marine invertebrate communities, as they provide predictable stable habitats and nutrients [96,107]. Thus, they provide predictable resources for marine predators, but also bathymetric features that can be used as cues for resource availability and aids for navigation which can be memorized [32,49]. This theory is supported by a high level of behavioural consistency displayed by other benthic foraging seabirds [79,108,109]. By contrast, the less consistent individuals in the present study foraged in pelagic habitats (PD-pelagic and PE-pelagic) and were likely to exploit schooling fish [49], a temporally and spatially variable prey resource, influenced by various oceanographic processes subject to intra- and inter-annual variations affecting prey species in the region [110].

The differences in consistency observed between individuals foraging in inshore and pelagic environments is unlikely to be due to the geometry, size or complexity of the available habitat. Despite the narrowness of the northwest/southeast axis of continental shelf habitat frequented by PD-pelagic animals, these individuals displayed less consistency in bearing than those foraging in the more circular PPB area. Similarly, despite the arc of available headings to potential foraging areas being similar for the PE-pelagic and PE-inshore birds, the later were more consistent in their bearings from the colony. In addition, while the area of Bass Strait used by PE-pelagic birds was approximately two to three times the area of PBB, PE-inshore birds represented less than 20% of the sampled population, such that the latter are likely to have had greater *per capita* available habitat yet displayed more consistency in their foraging behaviour. Furthermore, despite PBB representing a smaller absolute area, it is characterized by greater habitat diversity (seagrass beds, rock reefs, shallow sand-banks and deeper channels) compared to the open water habitat of Bass Strait.

Finally, some individuals from PE in the present study adopted a strategy of consistently foraging in both pelagic (Bass Strait) and inshore (PPB) habitats, either within the same or successive foraging trips, suggesting a degree of behavioural plasticity. Similar findings have been reported for gentoo penguins (*Pygoscelis papua*) in which some individuals switched between pelagic and benthic strategies on successive foraging trips [111]. While it is not known whether this mixed foraging strategy has specific benefits, it has been suggested that spatial and temporal environmental variation and resource competition can promote different adaptive responses in individuals, giving rise to different levels of plasticity [112].

# 5. Conclusion

In summary, the present study found foraging behaviour in Australasian gannets to be influenced primarily by colony, breeding stage and year, reflecting the spatial and temporal variation in resources around breeding colonies and, to a lesser degree, by sex. Overall, low to moderate levels of behavioural consistency were observed, decreasing with increasing timescales between sampling, but higher in inshore environments, where individuals displayed more benthic foraging strategies associated with more stable and predictable environments. These findings could have important implications for population dynamics as individuals may not be uniformly affected by environmental variability. Southeastern Australia is one of the fastest warming marine areas in the world and the anticipated oceanographic changes are likely to affect the distribution, abundance and diversity of prey species [41,42]. Inter-individual differences in foraging behaviour and behavioural plasticity in Australasian gannets, therefore, could affect how the population responds to changing environmental conditions [113]. Future studies should investigate the links between specific environmental conditions

and behavioural consistency further, and the benefits conferred by strategies adopted by individuals and breeding pairs on reproductive success.

Ethics. All procedures were conducted under the approval of the Deakin University Animal Ethics Committee (B20–2013), the Department of Environment, Land, Water and Planning (DELWP, Wildlife Research Permit No. 10006878, File No. FF383295) of the Victorian Government and the Australian Bird and Bat Banding Scheme (ABBBS, R Class Banding Licence No. 3168). All applicable institutional and/or national guidelines for the care and use of animals were followed.

Data accessibility. See the electronic supplementary material to obtain data used in this paper.

Authors' contributions. M.A.R.-M. and J.P.Y.A. conceived and designed the study. M.A.R.-M. collected the data. M.A.R.-M., E.C.M.C. and L.P.A. performed and contributed to the data analysis. M.A.R.-M. and J.P.Y.A. wrote the paper. All authors gave final approval for publication.

Competing interests. The authors of this manuscript declare no competing interests.

Funding. This research was partly funded by the Holsworth Wildlife Research Endowment and Birdlife Australia.

Acknowledgements. We thank the generous support from the Victorian Marine Science Consortium (especially to Roderick Watson, Elizabeth McGrath and Yvonne Gilbert), Parks Victoria and the Point Danger Committee of Management (Ewen Lovell and Phillip King) for facilitating the field activities required for this study. The assistance of numerous volunteers who participated in field activities is gratefully acknowledged, as is the advice on statistical analyses provided by Dr Peter Biro from Deakin University.

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
