## [Reviewer comments · Royal Society Open Science]

Review History

RSOS-181423.R0 (Original submission)

Review form: Reviewer 1

Is the manuscript scientifically sound in its present form?

No

Are the interpretations and conclusions justified by the results?

No

Is the language acceptable?

Yes

Is it clear how to access all supporting data?

No

Do you have any ethical concerns with this paper?

No

Have you any concerns about statistical analyses in this paper?

No

Recommendation?

Major revision is needed (please make suggestions in comments)

Comments to the Author(s)

The authors present a case study of fidelity in a near-shore foraging seabird between colonies, two years, and within breeding seasons. I found the introduction, methods and results sections relatively easy to read, however, the discussion was hard to follow and lacked in-depth interpretation of the results.

The explanation of the what timeframe the models were tested under can very late in the methods and thus it was not clear in all the models previously described which timeframes were being used.

Assigning a mixed foraging strategy to individuals that switched from benthic to pelagic habitats seems to inflate the degree of foraging fidelity observed and care should be taken to justify this choice and better present the amount of switching between habitats if this category is retained in the analysis. Is a mixed strategy is more likely to be assigned if an individual is tracked for more trips?

The supplementary materials do not include the data and codes used in this paper.

See a number of minor comments below. The line numbers in the manuscript do not align with the lines of text and a repeated for each page.

P1L40: extrinsic factors such as geographic location, local weather or predation risk [4-6] - distribution of prey should be included as it is important for foraging.

P1L49: "different timescales" -jumping from evolutionary time to implying how a two-year study might fill the gap on "different timescales" is a big leap.

P2L2-3: the foraging ecology of marine life is highly influenced by its environmental phenomena [20] - this is unclear, what is an environmental phenomenon?

P2L3: At a global scale, oceans display clear patterns of water circulation and climate [21]. - arguably at small spatial scales both circulation and climate are apparent. I don't see how this contrast helps to explain the motivations for your study. Clarify or remove.

P2L4: At local scales, physical features such as bathymetry, tidal regimes and nutrient fluxes determine the structure of marine and coastal ecosystems and influence the behavior of marine fauna. - bathymetry determines the structure of ecosystems at global scales as well. It doesn't seem important to disentangle spatial scales here as the birds are operating on small spatial scales.

P2L5-9: So upper-trophic level organisms should be more consistent than lower trophic levels? Does this make sense for krill, forage fish etc? It seems like krill might have a limited diversity of foraging strategies. Rephrase and be more specific.

P2L14: These attributes have ...stage of the annual cycle - you are already limiting your discussion to periods of central place foraging. Rephrase.

P2L15: Consider replacing "develop" with "exhibit".

P2L20: Italicize species name.

P2L24: Expansions in ranges or shifts? The title of the citation implies shifts.

P2L26: anticipated changes - Haven't the range expansions/shifts already occurred? Clarify the context of your study in relation to the changes discussed above.

P2L35: Be more specific about "timescales" as you have a two-year study with some between year and within year comparisons.

P2L45/L48: Delete "Previous studies have shown that" and diversify the sentence beginning so you aren't using "individuals from this colony" twice in a row.

P2L48: delete the word "may"

P2L58: How were accelerometer placements standardized between birds? Did the accelerometer always go on the bottom of the package, or?

P3L6: The same individuals or different individuals?

P3L11: Deployment data were split into individual foraging trips by visual inspection of the raw GPS tracks. - Using R as stated in the first sentence and set distance to the colony, and/or a number of locations, and other appropriate filters to identify foraging trips is a more standard methodology! Visual inspection to split foraging trips is not a repeatable method and is subjective.

P3L18-24: Were the manually tagged accelerometry profiles used to train the k-means clustering? It says unsupervised so I suspect not. Why manually tag the accelerometry data? Clarify this.

P3L22-24: Was the average VeDBA calculated for each trip or per activity? Clarify.

P3L29: Replace estimated with calculated.

P3L36: As a focus of the present study was to investigate broad scale temporal and geographic influences on foraging behavior. "Broad-scale"? Rephrase this sentence to be consistent with the introduction.

P3L40-41: Of the model residuals?

P3L40-48: Add a sentence or phrases justifying why this statistical approach was chosen for these data. Specifically, address why dredging was used for identifying the best-fixed structure and model averaging was used.

P3L58-P4L3: This paragraph should come sooner as now I don't understand what temporal comparisons the previous models were using. Are they the same models as these?

P4L24: Are these results found in a table or figure? Something should be referenced here.

P4L24: A topic sentence that reflects the motivation for these statistics would be helpful here.

P4L41: How were classifications controlled for by the number of trips sampled for each individual? Perhaps longer deployments resulted in more mixed classifications? The mixed classification also seems like a way to artificially inflate consistency in the foraging metrics. This should be strongly/more clearly justified or removed from the analysis.

P5L4-6: A stronger 1st sentence of the discussion should be considered since the current one sheds no light on the interpretation, implications, or importance of the study.

P5L9-11: This seems like an oversimplification due to the omission of the mixed strategy birds.

P5L12-14: This statement is repeated almost verbatim from the methods and points to an additional analysis that could be done with these data. It reads like a disclaimer. Move this somewhere lower in the discussion and provide context by suggesting what other environmental variables might be useful for this next step. Why would adding more environmental variables be useful? Given your results how would you approach this? Also, conclude this paragraph with a statement that summarizes the most important findings of this study.

P5L17-18: Given your topic sentence I thought you were going to justify the use of metrics in ecology. Rewrite to provide context for geographic variation.

P5L30: Omit the word “interestingly”. If you are discussing something it should be interesting. Rephrase the topic sentence of this paragraph to discuss links between foraging and breeding success.

P5L39. Omit “In the present study” and rewrite topic sentence.

P5L39: How do your results show “temporal variation” in relation to the breeding stage? You are discussing differences between breeding stages, but “temporal variation” as mentioned here is very vague. Rephrase your topic sentence to discuss the importance of changes in foraging patterns during different reproductive stages.

P5L49-57: Repeated paragraph.

P6L1-8: How does the repeatability of VeDBA factor into this discussion?

P6L10-14: Integrate this discussion topic with other results.

Review form: Reviewer 2

Is the manuscript scientifically sound in its present form?

No

Are the interpretations and conclusions justified by the results?

No

Is the language acceptable?

Yes

Is it clear how to access all supporting data?

No

Do you have any ethical concerns with this paper?

No

Have you any concerns about statistical analyses in this paper?

Yes

Recommendation?

Major revision is needed (please make suggestions in comments)

Comments to the Author(s)

General comments:

The manuscript titled “Influences on foraging behavior and consistency in a marine aerial predator” raises some interesting questions, but the study only goes part of the way to addressing them. Specifically, the Introduction raises interesting and significant ecological questions about the relationship between behavioral consistency, individual specialization, and the potential for adaptation in the context of climate change. The Introduction also alludes to the importance of understanding behavioral consistency as a basis for predictions about responses to environmental change. However, the study presented here is confined to a statistical analysis of the relationship between the foraging metrics of Australasian gannets (*Morus serrator*), fixed effects such as year, colony, pelagic/nearshore habitat, sex, breeding stage, and random effects at the nest and individual levels. The study does not incorporate any environmental factors that might explain variation in foraging metrics, and relies heavily on previous studies for interpretation. The main inference is that the apparent significance of individual random effects implies individual specialization and behavioral consistency. The main difference from similar studies of foraging metrics is that the same individuals were tagged on several occasions, allowing for comparisons between trips, breeding stage, and year that may provide information on the decay in behavioral consistency over time. The final conclusion that behavioral consistency could have important implications for population dynamics may well be true, but is based on the first two paragraphs of the Introduction rather than the study presented here. Similar questions about ecological traps due to behavioral consistency were raised and addressed by Sherley et al. *Curr. Biol.* 2017, but that paper is not cited here. Overall, the study falls short of the expectations raised in the Introduction. While it is useful to set out the broad ecological questions impression, it is also important to be clear about the specific question you will address in your study and what we will know at the end of the paper that was not known before.

Major comments:

My main methodological concern is with the removal of ‘erroneous locations’ based on a speed filter of 55 km/h. GPS is usually assumed to be sufficiently accurate that filtering of data points is rarely necessary. The implementation of the filter is unclear – the text suggests you removed any points that implied a speed of greater than 55 km/h, but if 55 km/h is an average speed, then speeds will sometimes be faster. Removal of datapoints will affect the distance traveled and the tortuosity index, so it becomes unclear whether differences in these metrics reflect genuine differences in foraging patterns or variation in GPS performance.

It would be useful to see the delta-AICs between the selected fixed effects model and a mixed effects model with similar fixed structure for each foraging metric to confirm that the individual random effects are indeed supported by the data.

Given that your question concerns the additional explanatory power of individual random effects, it’s unclear whether model averaging is appropriate – why not simply use the fixed model with lowest AICc?

In your second set of models, you are treating the output from the first set of models as data (P. 3; L 54). This is questionable without a clear method for propagating uncertainty.

What do you mean by “The dataset acquired was then modified ...” (P. 4; L 3)?

You report a consistent decrease in variance associated with individual random effects over time (P. 4; LL 37-39), but how can we assess whether the reported decrease was significant? If this is your main result, it needs clearer support from the data and analysis.

Minor comments:

P. 1; L.34: Result here appears to differ from results presented on P 4. L 39.

P. 2; L 25 ff: The scope of work indicated here is far greater than achieved by this study.

P 3; LL 6-7: Why?

P 5; L 49 ff: Duplicated paragraph.

P 5; L 23: What do you mean by "habitat accessibility" here and elsewhere?

P 6; L 6: How are you measuring dive rate and VeDBA here? If you are looking at number of dives per trip and average VeDBA over the course of a trip, then it would not be surprising that average VeDBA is lower on longer trips, despite more dives (see Boyd et al. Ecol Model. 2015). If, on the other hand, you are referring to total VeDBA per average trip, then yes it would be very surprising to find this is lower for larger-bodied animals on longer trips with more dives, and this would demand further examination.

P 6; L 13: I think you mean foraged in both habitats rather than displayed both behaviors. If not, how do you know whether they foraged alone or in a group?

Decision letter (RSOS-181423.R0)

18-Dec-2018

Dear Ms Rodríguez Malagon,

The editors assigned to your paper ("Influences on foraging behaviour and consistency in a marine aerial predator") have now received comments from reviewers. We would like you to revise your paper in accordance with the referee and Associate Editor suggestions which can be found below (not including confidential reports to the Editor). Please note this decision does not guarantee eventual acceptance.

Please submit a copy of your revised paper before 10-Jan-2019. Please note that the revision deadline will expire at 00.00am on this date. If we do not hear from you within this time then it will be assumed that the paper has been withdrawn. In exceptional circumstances, extensions may be possible if agreed with the Editorial Office in advance. We do not allow multiple rounds of revision so we urge you to make every effort to fully address all of the comments at this stage. If deemed necessary by the Editors, your manuscript will be sent back to one or more of the original reviewers for assessment. If the original reviewers are not available, we may invite new reviewers.

If your study uses humans or animals please include details of the ethical approval received, including the name of the committee that granted approval. For human studies please also detail

whether informed consent was obtained. For field studies on animals please include details of all permissions, licences and/or approvals granted to carry out the fieldwork.

- Data accessibility

If you wish to submit your supporting data or code to Dryad (<http://datadryad.org/>), or modify your current submission to dryad, please use the following link:
<http://datadryad.org/submit?journalID=RSOS&manu=RSOS-181423>

- Competing interests

- Authors' contributions

- Acknowledgements

- Funding statement

Please note that Royal Society Open Science charge article processing charges for all new submissions that are accepted for publication. Charges will also apply to papers transferred to Royal Society Open Science from other Royal Society Publishing journals, as well as papers submitted as part of our collaboration with the Royal Society of Chemistry (<http://rsos.royalsocietypublishing.org/chemistry>). If your manuscript is newly submitted and subsequently accepted for publication, you will be asked to pay the article processing charge, unless you request a waiver and this is approved by Royal Society Publishing. You can find out more about the charges at <http://rsos.royalsocietypublishing.org/page/charges>. Should you have any queries, please contact openscience@royalsociety.org.

on behalf of Dr Ari Friedlaender (Associate Editor) and Kevin Padian (Subject Editor)
 openscience@royalsociety.org

Comments to Author:

Reviewers' Comments to Author:
 Reviewer: 1

Comments to the Author(s)

The authors present a case study of fidelity in a near-shore foraging seabird between colonies, two years, and within breeding seasons. I found the introduction, methods and results sections relatively easy to read, however, the discussion was hard to follow and lacked in-depth interpretation of the results.

The explanation of the what timeframe the models were tested under can very late in the methods and thus it was not clear in all the models previously described which timeframes were being used.

Assigning a mixed foraging strategy to individuals that switched from benthic to pelagic habitats seems to inflate the degree of foraging fidelity observed and care should be taken to justify this choice and better present the amount of switching between habitats if this category is retained in the analysis. Is a mixed strategy is more likely to be assigned if an individual is tracked for more trips?

The supplementary materials do not include the data and codes used in this paper.

See a number of minor comments below. The line numbers in the manuscript do not align with the lines of text and a repeated for each page.

P1L40: extrinsic factors such as geographic location, local weather or predation risk [4-6] - distribution of prey should be included as it is important for foraging.

P1L49: "different timescales" -jumping from evolutionary time to implying how a two-year study might fill the gap on "different timescales" is a big leap.

P2L2-3: the foraging ecology of marine life is highly influenced by its environmental phenomena [20] - this is unclear, what is an environmental phenomenon?

P2L3: At a global scale, oceans display clear patterns of water circulation and climate [21]. - arguably at small spatial scales both circulation and climate are apparent. I don't see how this contrast helps to explain the motivations for your study. Clarify or remove.

P2L4: At local scales, physical features such as bathymetry, tidal regimes and nutrient fluxes determine the structure of marine and coastal ecosystems and influence the behavior of marine

fauna. – bathymetry determines the structure of ecosystems at global scales as well. It doesn't seem important to disentangle spatial scales here as the birds are operating on small spatial scales.

P2L5-9: So upper-trophic level organisms should be more consistent than lower trophic levels? Does this make sense for krill, forage fish etc? It seems like krill might have a limited diversity of foraging strategies. Rephrase and be more specific.

P2L14: These attributes havestage of the annual cycle – you are already limiting your discussion to periods of central place foraging. Rephrase.

P2L15: Consider replacing “develop” with “exhibit”.

P2L20: Italicize species name.

P2L24: Expansions in ranges or shifts? The title of the citation implies shifts.

P2L26: anticipated changes - Haven't the range expansions/shifts already occurred? Clarify the context of your study in relation to the changes discussed above.

P2L35: Be more specific about “timescales” as you have a two-year study with some between year and within year comparisons.

P2L45/L48: Delete “Previous studies have shown that” and diversify the sentence beginning so you aren't using “individuals from this colony” twice in a row.

P2L48: delete the word “may”

P2L58: How were accelerometer placements standardized between birds? Did the accelerometer always go on the bottom of the package, or?

P3L6: The same individuals or different individuals?

P3L11: Deployment data were split into individual foraging trips by visual inspection of the raw GPS tracks. – Using R as stated in the first sentence and set distance to the colony, and/or a number of locations, and other appropriate filters to identify foraging trips is a more standard methodology! Visual inspection to split foraging trips is not a repeatable method and is subjective.

P3L18-24: Were the manually tagged accelerometry profiles used to train the k-means clustering? It says unsupervised so I suspect not. Why manually tag the accelerometry data? Clarify this.

P3L22-24: Was the average VeDBA calculated for each trip or per activity? Clarify.

P3L29: Replace estimated with calculated.

P3L36: As a focus of the present study was to investigate broad scale temporal and geographic influences on foraging behavior. “Broad-scale”? Rephrase this sentence to be consistent with the introduction.

P3L40-41: Of the model residuals?

P3L40-48: Add a sentence or phrases justifying why this statistical approach was chosen for these data. Specifically, address why dredging was used for identifying the best-fixed structure and model averaging was used.

P3L58-P4L3: This paragraph should come sooner as now I don't understand what temporal comparisons the previous models were using. Are they the same models as these?

P4L24: Are these results found in a table or figure? Something should be referenced here.

P4L24: A topic sentence that reflects the motivation for these statistics would be helpful here.

P4L41: How were classifications controlled for by the number of trips sampled for each individual? Perhaps longer deployments resulted in more mixed classifications? The mixed classification also seems like a way to artificially inflate consistency in the foraging metrics. This should be strongly/more clearly justified or removed from the analysis.

P5L4-6: A stronger 1st sentence of the discussion should be considered since the current one sheds no light on the interpretation, implications, or importance of the study.

P5L9-11: This seems like an oversimplification due to the omission of the mixed strategy birds.

P5L12-14: This statement is repeated almost verbatim from the methods and points to an additional analysis that could be done with these data. It reads like a disclaimer. Move this somewhere lower in the discussion and provide context by suggesting what other environmental variables might be useful for this next step. Why would adding more environmental variables be useful? Given your results how would you approach this? Also, conclude this paragraph with a statement that summarizes the most important findings of this study.

P5L17-18: Given your topic sentence I thought you were going to justify the use of metrics in ecology. Rewrite to provide context for geographic variation.

P5L30: Omit the word "interestingly". If you are discussing something it should be interesting. Rephrase the topic sentence of this paragraph to discuss links between foraging and breeding success.

P5L39. Omit "In the present study" and rewrite topic sentence.

P5L39: How do your results show "temporal variation" in relation to the breeding stage? You are discussing differences between breeding stages, but "temporal variation" as mentioned here is very vague. Rephrase your topic sentence to discuss the importance of changes in foraging patterns during different reproductive stages.

P5L49-57: Repeated paragraph.

P6L1-8: How does the repeatability of VeDBA factor into this discussion?

P6L10-14: Integrate this discussion topic with other results.

Reviewer: 2

Comments to the Author(s)

General comments:

The manuscript titled "Influences on foraging behavior and consistency in a marine aerial predator" raises some interesting questions, but the study only goes part of the way to addressing them. Specifically, the Introduction raises interesting and significant ecological questions about the relationship between behavioral consistency, individual specialization, and the potential for adaptation in the context of climate change. The Introduction also alludes to the

importance of understanding behavioral consistency as a basis for predictions about responses to environmental change. However, the study presented here is confined to a statistical analysis of the relationship between the foraging metrics of Australasian gannets (*Morus serrator*), fixed effects such as year, colony, pelagic/nearshore habitat, sex, breeding stage, and random effects at the nest and individual levels. The study does not incorporate any environmental factors that might explain variation in foraging metrics, and relies heavily on previous studies for interpretation. The main inference is that the apparent significance of individual random effects implies individual specialization and behavioral consistency. The main difference from similar studies of foraging metrics is that the same individuals were tagged on several occasions, allowing for comparisons between trips, breeding stage, and year that may provide information on the decay in behavioral consistency over time. The final conclusion that behavioral consistency could have important implications for population dynamics may well be true, but is based on the first two paragraphs of the Introduction rather than the study presented here. Similar questions about ecological traps due to behavioral consistency were raised and addressed by Sherley et al. *Curr. Biol.* 2017, but that paper is not cited here. Overall, the study falls short of the expectations raised in the Introduction. While it is useful to set out the broad ecological questions impression, it is also important to be clear about the specific question you will address in your study and what we will know at the end of the paper that was not known before.

Major comments:

My main methodological concern is with the removal of ‘erroneous locations’ based on a speed filter of 55 km/h. GPS is usually assumed to be sufficiently accurate that filtering of data points is rarely necessary. The implementation of the filter is unclear – the text suggests you removed any points that implied a speed of greater than 55 km/h, but if 55 km/h is an average speed, then speeds will sometimes be faster. Removal of datapoints will affect the distance traveled and the tortuosity index, so it becomes unclear whether differences in these metrics reflect genuine differences in foraging patterns or variation in GPS performance.

It would be useful to see the delta-AICs between the selected fixed effects model and a mixed effects model with similar fixed structure for each foraging metric to confirm that the individual random effects are indeed supported by the data.

Given that your question concerns the additional explanatory power of individual random effects, it’s unclear whether model averaging is appropriate – why not simply use the fixed model with lowest AICc?

In your second set of models, you are treating the output from the first set of models as data (P. 3; L 54). This is questionable without a clear method for propagating uncertainty.

What do you mean by “The dataset acquired was then modified ...” (P. 4; L 3)?

You report a consistent decrease in variance associated with individual random effects over time (P. 4; LL 37-39), but how can we assess whether the reported decrease was significant? If this is your main result, it needs clearer support from the data and analysis.

Minor comments:

P. 1; L.34: Result here appears to differ from results presented on P 4. L 39.

P. 2; L 25 ff: The scope of work indicated here is far greater than achieved by this study.

P 3; LL 6-7: Why?

P 5; L 49 ff: Duplicated paragraph.

P 5; L 23: What do you mean by “habitat accessibility” here and elsewhere?

P 6; L 6: How are you measuring dive rate and VeDBA here? If you are looking at number of dives per trip and average VeDBA over the course of a trip, then it would not be surprising that average VeDBA is lower on longer trips, despite more dives (see Boyd et al. *Ecol Model.* 2015). If, on the other hand, you are referring to total VeDBA per average trip, then yes it would be very

surprising to find this is lower for larger-bodied animals on longer trips with more dives, and this would demand further examination.

P 6; L 13: I think you mean foraged in both habitats rather than displayed both behaviors. If not, how do you know whether they foraged alone or in a group?

Author's Response to Decision Letter for (RSOS-181423.R0)

See Appendix A.

RSOS-181423.R1 (Revision)

Review form: Reviewer 3

Is the manuscript scientifically sound in its present form?

No

Are the interpretations and conclusions justified by the results?

No

Is the language acceptable?

Yes

Do you have any ethical concerns with this paper?

No

Have you any concerns about statistical analyses in this paper?

Yes

Recommendation?

Accept with minor revision (please list in comments)

Comments to the Author(s)

The manuscript is well written and tackles some very interesting ecological questions. While the argument that foraging behaviour will be more consistent in more stable environments is convincing, it is not clear that the variability in behaviour resulting from increased environmental stability can be detected from tracking data. The data collection methods, authors' approach to quantifying and measuring the consistency of foraging behaviour are both thorough and appropriate.

My great concern is whether the results of the manuscript might be explained purely by the geometry of the habitat. The authors' analyses suggest that foraging behaviour is more consistent in benthic habitats than pelagic habitats, but to what degree are these results dependent on the geometry of the foraging region?

Consider the bearing metric and suppose the foraging region is a thin, elongated ellipse. If the colony is situated along the semi-major axis of the ellipse then the bearing will be highly constrained, but if the colony is situated on the semi-minor axis the bearing can be more variable. The authors need to discuss this issue before the manuscript is suitable for publication.

In addition, there are some minor issues in relation to the choice and description of the methods.

As noted on P26L54, bearing is a circular variable. So standard modelling techniques will not be valid if the bearings encountered in practice span the 0/360 degree boundary. There should be some justification of the use of standard regression techniques for bearing.

P26L1 It is the clustering that is unsupervised, not the wavelet transformation. I presume the intention was "The Ethnographer package was used to identify these behaviours by performing a k-means, unsupervised cluster analyses of one second windows of continuous wavelet spectra computed from the time series."

Review form: Reviewer 4

Is the manuscript scientifically sound in its present form?

Yes

Are the interpretations and conclusions justified by the results?

Yes

Is the language acceptable?

Yes

Do you have any ethical concerns with this paper?

No

Have you any concerns about statistical analyses in this paper?

No

Recommendation?

Major revision is needed (please make suggestions in comments)

Comments to the Author(s)

Ms. No. RSOS-181423.R1

"Factors influencing foraging behavior and consistency in a marine aerial predator"

Overall, the authors have done a nice work, revising the manuscript in accordance with the reviewer's comment. However, there is one critical concern regarding the interpretation of the results.

The authors state that the inshore foraging (PPB) exhibited more consistency, compared to the pelagic and mixed, in the foraging metrics, probably relating to its stability and predictability. However, it could be also because condition in foraging habitats in PPB is much less varied than these in Bass Strait (i.e. much smaller area to forage and narrower range of directions to travel in PPB: Fig. 1), which should restrict birds to occupy only specific habitats. The authors are probably true, in some parts, explaining that "seagrasses and sandy bottoms provide predictable stable habitats and features that can be used as cues for resource availability", but I also believe that comparing consistency in areas between LARGE and SMALL habitats is not fare to do (availability in habitat choice is very different). Please mention or explain this point in the Discussion. Otherwise, this study would be just to examine differences in foraging behavior of Australian gannets between sexes, breeding-stages, colonies, and years, which is not new and probably does not match to the description in the Introduction.

Minor comment:

“the average Vectorial Dynamic Body Acceleration (VeDBA) per activity and then all activity values were averaged within each foraging trip” (P4L4–6 in the cleared version of the revised ms).

> Why did you use the “averaged” values rather than the “sum” of VeDBA?

> Do you mean “per activity” as “per diving”?

Decision letter (RSOS-181423.R1)

17-Jan-2020

Dear Dr Arnould:

Manuscript ID RSOS-181423.R1 entitled "Factors influencing foraging behaviour and consistency in a marine aerial predator" which you submitted to Royal Society Open Science, has been reviewed. The comments of the reviewer(s) are included at the bottom of this letter.

Please submit a copy of your revised paper before 09-Feb-2020. Please note that the revision deadline will expire at 00.00am on this date. If we do not hear from you within this time then it will be assumed that the paper has been withdrawn. In exceptional circumstances, extensions may be possible if agreed with the Editorial Office in advance. We do not allow multiple rounds of revision so we urge you to make every effort to fully address all of the comments at this stage. If deemed necessary by the Editors, your manuscript will be sent back to one or more of the original reviewers for assessment. If the original reviewers are not available we may invite new reviewers.

- Ethics statement

- Data accessibility

It is a condition of publication that all supporting data are made available either as supplementary information or preferably in a suitable permanent repository. The data accessibility section should state where the article's supporting data can be accessed. This section should also include details, where possible of where to access other relevant research materials

such as statistical tools, protocols, software etc can be accessed. If the data have been deposited in an external repository this section should list the database, accession number and link to the DOI for all data from the article that have been made publicly available. Data sets that have been deposited in an external repository and have a DOI should also be appropriately cited in the manuscript and included in the reference list.

- Competing interests

- Authors' contributions

- Acknowledgements

- Funding statement

Kind regards,

Andrew Dunn

on behalf of Dr Ari Friedlaender (Associate Editor) and Kevin Padian (Subject Editor)

Editor comments:

The authors have done a nice job of responding to reviewer comments in their revised submission. However, the two current reviewers raise additional questions that appear non-trivial. Some of the modifications suggested require re-analysis and/or revision of the conclusions. The data presented are valuable, but as one reviewer notes, without some clarification and more meaningful discussion the value of the results in terms of new information would be limited. The second reviewer also notes difficulty with accepting geometry of the

habitat as a main factor; with a sample size of two, this could be a point well taken and I hope you will address this in your final revision.

I also suggest that you change your title to note "the Australian gannet" instead of "a marine aerial predator," which is vaguer. You might also specify the "factors" in the title so that readers know what to expect. Thanks and best wishes for your revision.

Reviewer comments to Author:

Reviewer: 3

Comments to the Author(s)

The manuscript is well written and tackles some very interesting ecological questions. While the argument that foraging behaviour will be more consistent in more stable environments is convincing, it is not clear that the variability in behaviour resulting from increased environmental stability can be detected from tracking data. The data collection methods, authors' approach to quantifying and measuring the consistency of foraging behaviour are both thorough and appropriate.

My great concern is whether the results of the manuscript might be explained purely by the geometry of the habitat. The authors' analyses suggest that foraging behaviour is more consistent in benthic habitats than pelagic habitats, but to what degree are these results dependent on the geometry of the foraging region?

Consider the bearing metric and suppose the foraging region is a thin, elongated ellipse. If the colony is situated along the semi-major axis of the ellipse then the bearing will be highly constrained, but if the colony is situated on the semi-minor axis the bearing can be more variable. The authors need to discuss this issue before the manuscript is suitable for publication.

In addition, there are some minor issues in relation to the choice and description of the methods.

As noted on P26L54, bearing is a circular variable. So standard modelling techniques will not be valid if the bearings encountered in practice span the 0/360 degree boundary. There should be some justification of the use of standard regression techniques for bearing.

P26L1 It is the clustering that is unsupervised, not the wavelet transformation. I presume the intention was "The Ethnographer package was used to identify these behaviours by performing a k-means, unsupervised cluster analyses of one second windows of continuous wavelet spectra computed from the time series."

Reviewer: 4

Comments to the Author(s)

Ms. No. RSOS-181423.R1

"Factors influencing foraging behavior and consistency in a marine aerial predator"

Overall, the authors have done a nice work, revising the manuscript in accordance with the reviewer's comment. However, there is one critical concern regarding the interpretation of the results.

The authors state that the inshore foraging (PPB) exhibited more consistency, compared to the pelagic and mixed, in the foraging metrics, probably relating to its stability and predictability. However, it could be also because condition in foraging habitats in PPB is much less varied than these in Bass Strait (i.e. much smaller area to forage and narrower range of directions to travel in PPB: Fig. 1), which should restrict birds to occupy only specific habitats. The authors are probably true, in some parts, explaining that "seagrasses and sandy bottoms provide predictable stable habitats and features that can be used as cues for resource availability", but I also believe that

comparing consistency in areas between LARGE and SMALL habitats is not fare to do (availability in habitat choice is very different). Please mention or explain this point in the Discussion. Otherwise, this study would be just to examine differences in foraging behavior of Australian gannets between sexes, breeding-stages, colonies, and years, which is not new and probably does not match to the description in the Introduction.

Minor comment:

“the average Vectorial Dynamic Body Acceleration (VeDBA) per activity and then all activity values were averaged within each foraging trip” (P4L4-6 in the cleared version of the revised ms).

> Why did you use the “averaged” values rather than the “sum” of VeDBA?

> Do you mean “per activity” as “per diving”?

Author's Response to Decision Letter for (RSOS-181423.R1)

See Appendix B.

RSOS-181423.R2 (Revision)

Review form: Reviewer 4

Is the manuscript scientifically sound in its present form?

Yes

Are the interpretations and conclusions justified by the results?

Yes

Is the language acceptable?

Yes

Do you have any ethical concerns with this paper?

No

Have you any concerns about statistical analyses in this paper?

No

Recommendation?

Accept as is

Comments to the Author(s)

Overall, the manuscript has been much simplified, discussion is clear, and the interpretation of results is adequate and acceptable. I believe the authors have done a great job on revising the manuscript. This is a nice and interesting study.

Decision letter (RSOS-181423.R2)

Dear Dr Arnould,

It is a pleasure to accept your manuscript entitled "Geographic, temporal and individual factors influencing foraging behaviour and consistency in Australasian gannets" in its current form for publication in Royal Society Open Science. The comments of the reviewer(s) who reviewed your manuscript are included at the foot of this letter.

on behalf of Dr Ari Friedlaender (Associate Editor) and Kevin Padian (Subject Editor)
openscience@royalsociety.org

Reviewer comments to Author:
Reviewer: 4

Comments to the Author(s)

Overall, the manuscript has been much simplified, discussion is clear, and the interpretation of results is adequate and acceptable. I believe the authors have done a great job on revising the manuscript. This is a nice and interesting study.

Appendix A

Ref.: Ms. No. RSOS-181423.R1

Factors influencing foraging behaviour and consistency in a marine aerial predator

Dear Reviewers and Editors,

We would like to use this opportunity to sincerely thank the reviewers for their detailed comments. They were all pertinent and useful for the improving of the manuscript. We have taken all these comments into consideration within the revised version. The corresponding changes and refinements made after revision are summarized in the following document 'Response to Referees'. Please find the editors and reviewers' comments in black and our answers are in blue.

Kind regards,

John Arnould (corresponding author)

Decision Letter (RSOS-181423)

From: openscience@royalsociety.org

To: marodrig@deakin.edu.au

CC: marodrig@deakin.edu.au, elodie.camprasse@deakin.edu.au, lauren.angel@deakin.edu.au, john.arnould@deakin.edu.au, journal-submit@datadryad.org

Subject: Royal Society Open Science - Decision on Manuscript ID RSOS-181423

Body:

18-Dec-2018

Dear Ms Rodríguez Malagon,

The editors assigned to your paper ("Influences on foraging behaviour and consistency in a marine aerial predator") have now received comments from reviewers. **We would like you to revise your paper in accordance with the referee and Associate Editor suggestions which can be found below** (not including confidential reports to the Editor). Please note this decision does not guarantee eventual acceptance.

Please submit a copy of your revised paper before 10-Jan-2019. Please note that the revision deadline will expire at 00.00am on this date. If we do not hear from you within this time then it will be assumed that the paper has been withdrawn. In exceptional circumstances, extensions may be possible if agreed with the Editorial Office in advance. We do not allow multiple rounds of revision so we urge you to make every effort to fully address all of the comments at this stage. If deemed necessary by the Editors, your manuscript will be sent back to one or more of the original reviewers for assessment. If the original reviewers are not available, we may invite new reviewers.

When submitting your revised manuscript, you must respond to the comments made by the referees and upload a file "**Response to Referees**" in "Section 6 - File Upload". Please use this to document how you have responded to the comments, and the adjustments you have made. In order to expedite the processing of the revised manuscript, please be as specific as possible in your response.

- Data accessibility

It is a condition of publication that all supporting data are made available either as supplementary information or preferably in a suitable permanent repository. The data accessibility section should state where the article's supporting data can be accessed. This section should also include details, where possible of where to access other relevant research materials such as statistical tools, protocols, software etc can be accessed. If the data have been deposited in an external repository

this section should list the database, accession number and link to the DOI for all data from the article that have been made publicly available. Data sets that have been deposited in an external repository and have a DOI should also be appropriately cited in the manuscript and included in the reference list.

<http://datadryad.org/submit?journalID=RSOS&manu=RSOS-181423>

- Competing interests

- Authors' contributions

All submissions, other than those with a single author, must include an Authors' Contributions section which individually lists the specific contribution of each author. The list of Authors should meet all of the following criteria; 1) substantial contributions to conception and design, or acquisition of data, or analysis and interpretation of data; 2) drafting the article or revising it critically for important intellectual content; and 3) final approval of the version to be published. All contributors who do not meet all of these criteria should be included in the acknowledgements.

- Acknowledgements

- Funding statement

Please note that Royal Society Open Science charge article processing charges for all new submissions that are accepted for publication. Charges will also apply to papers transferred to Royal Society Open Science from other Royal Society Publishing journals, as well as papers submitted as part of our collaboration with the Royal Society of Chemistry (<http://rsos.royalsocietypublishing.org/chemistry>). If your manuscript is newly submitted and subsequently accepted for publication, you will be asked to pay the article processing charge, unless you request a waiver and this is approved by Royal Society Publishing. You can find out more about the charges at <http://rsos.royalsocietypublishing.org/page/charges>. Should you have any queries, please contact openscience@royalsociety.org.

Kind regards,

Andrew Dunn

on behalf of Dr Ari Friedlaender (Associate Editor) and Kevin Padian (Subject Editor)

Reviewer #1:

The authors present a case study of fidelity in a near-shore foraging seabird between colonies, two years, and within breeding seasons. I found the introduction, methods and results sections relatively easy to read, however, the discussion was hard to follow and lacked in-depth interpretation of the results.

The discussion has now been rewritten to address these concerns.

The explanation of the what timeframe the models were tested under can very late in the methods and thus it was not clear in all the models previously described which timeframes were being used.

This has now been clarified at the end of the Introduction and in the Methods to avoid confusion.

Assigning a mixed foraging strategy to individuals that switched from benthic to pelagic habitats seems to inflate the degree of foraging fidelity observed and care should be taken to justify this choice and better present the amount of switching between habitats if this category is retained in the analysis. Is a mixed strategy is more likely to be assigned if an individual is tracked for more trips?

The mixed strategy we describe is not more likely to be assigned if an individual is tracked for more trips because this classification takes into account the number of trips obtained, and a minimum of three trips has been used for all birds. If a bird is classified as PE-inshore or PE-pelagic, it does not mean that no switching can happen, but rather that they still predominantly forage in one habitat or the other (proportion of all trips >70 %); otherwise, the individual is classified as having a mixed strategy (PE-mixed). This classification has been retained as one of our goals was to investigate the link between habitat selection and consistency and we are taking into account the number of trips obtained for each individual to avoid biases.

The supplementary materials do not include the data and codes used in this paper.

We will submit the data online. We did not include any code in the supplementary materials as we did not use any novel technique for the creation of this research. We rather used and followed recommended R packages/functions and statistical protocols published by other authors.

Researchers interested in replicating our methods should consult the papers we cited here and judge for themselves the convenience of using these in their own data in accordance with the recommendations of the cited authors and the relevant R package documentation (e.g *trip*, *adehabitatHR*, *nlme*, *ape*, *MuMIn*).

See a number of minor comments below.

The line numbers in the manuscript do not align with the lines of text and a repeated for each page.

Yes, the line numbers were created automatically by the uploading system of the journal, not by us. We apologise for the confusion this may have caused.

Specific comments

P2L14: These attributes havestage of the annual cycle – you are already limiting your discussion to periods of central place foraging. Rephrase.

“stage of the annual cycle” has been removed.

P2L15: Consider replacing “develop” with “exhibit”.

This word has been changed to "exhibit" as recommended.

P2L20: Italicize species name.

The species name has been italicized.

P2L24: Expansions in ranges or shifts? The title of the citation implies shifts.

Both expansions and shifts in marine species ranges have occurred (Johnson et al. 2011) and we have now changed the text to reflect this.

P2L26: anticipated changes - Haven't the range expansions/shifts already occurred? Clarify the context of your study in relation to the changes discussed above.

That is right. The text has been changed accordingly: “is necessary to understand how their populations may adapt to changes in the supply of marine resources” was added, and “is necessary to predict how their populations may respond to the anticipated changes” was erased.

P2L35: Be more specific about “timescales” as you have a two-year study with some between year and within year comparisons.

“timescales” was replaced by “within and between years”.

P2L45/L48: Delete “Previous studies have shown that” and diversify the sentence beginning so you aren't using “individuals from this colony” twice in a row.

“Previous studies have shown that” was erased in both sentences and “Birds from PE” was included in the second sentence.

P2L48: delete the word “may”

The word “may” has been deleted.

P2L58: How were accelerometer placements standardized between birds? Did the accelerometer always go on the bottom of the package, or?

Yes, each GPS was always on top of the package as it is slightly bigger and heavier. Accelerometer was always at the bottom and oriented in the same direction each time. This is a standardised method within our research lab and all deployments are done in the same way. This information now appears in the text.

P3L6: The same individuals or different individuals?

The “same individuals” was added to the text.

P3L11: Deployment data were split into individual foraging trips by visual inspection of the raw GPS tracks. – Using R as stated in the first sentence and set distance to the colony, and/or a number of locations, and other appropriate filters to identify foraging trips is a more standard methodology! Visual inspection to split foraging trips is not a repeatable method and is subjective.

We changed the text to reflect what we did, which is a visual inspection of the data before using the times birds returned to the colonies as an end points for the trips. Apologies for the confusion.

P3L18-24: Were the manually tagged accelerometry profiles used to train the k-means clustering? It says unsupervised so I suspect not. Why manually tag the accelerometry data? Clarify this.

Our paper says:

“Data were initially inspected visually to assign foraging behaviours (plunge diving and surface foraging) in IGOR Pro (Version 6.37, WaveMetrics, USA) [63], based on the acceleration profiles suggested for other species of gannets and boobies [64-66].”

This means the previous reported acceleration profiles were used to inspect the raw data in order to identify the behaviours of interest within them, but at this point they were only used as a reference.

Then:

“The Ethographer package was then used to perform a k-means clustering analysis to identify these behaviours using an unsupervised continuous wavelet transformation (1 s window) [67]. “

An unsupervised classification was used in all foraging trips where different behaviours were identified and later, each identified cluster was assigned a specific behaviour based on the previous visual identification. To clarify this point within the manuscript the following line was added to the text: “Later, each identified cluster was assigned with a specific behaviour based on the previous visual identification.”

P3L22-24: Was the average VeDBA calculated for each trip or per activity? Clarify.

VeDBA was estimated per activity, then all activity values were averaged within each foraging trip. This is now clarified in the text.

P3L29: Replace estimated with calculated.

The word “estimated” was replaced by “calculated”.

P3L36: As a focus of the present study was to investigate broad scale temporal and geographic influences on foraging behavior. “Broad-scale”? Rephrase this sentence to be consistent with the introduction.

Yes, thank you for your comment. This sentence has now been rephrased.

P3L40-41: Of the model residuals?

Yes, that is right. We have now clarified this in the text.

P3L40-48: Add a sentence or phrases justifying why this statistical approach was chosen for these data. Specifically, address why dredging was used for identifying the best-fixed structure and model averaging was used.

These lines were included in the text: “This multi-model statistical approach was selected as it allows to identify strong associations between multiple explanatory variables, while AIC values compare multiple models all at once incorporating model selection uncertainty and enabling inferences that are unconditional on a specific model [76]. In addition, through averaging two or more models with similar support levels, it is possible to obtain parameter estimates in a robust way [77].”

New reference included to support our arguments:

“[77] Grueber C, Nakagawa S, Laws R, Jamieson I (2011) Multimodel inference in ecology and evolution: challenges and solutions. *J Evol Biol* 24:699-711”

P3L58-P4L3: This paragraph should come sooner as now I don't understand what temporal comparisons the previous models were using. Are they the same models as these?

Yes, sorry for this confusion. To make methods clearer we added these lines into the text, right after the description of the fixed parameters used in the first set of models: "For these models the full dataset was used (i.e. data obtained from all individuals in both sites and two years of sampling) using sampling size of three or more foraging trips per deployment."

P4L24: Are these results found in a table or figure? Something should be referenced here.

All results presented in this paragraph (P4L24-30) are referenced from Table 2 and the Supplementary tables S1 and S2. The description of these results was based on the parameter estimates resulting from each model. The sign and value of each category of each fixed factor show its importance within the final model.

P4L24: A topic sentence that reflects the motivation for these statistics would be helpful here.

These lines were added into text: "The top-ranked statistical models explaining the factors influencing the foraging behaviour of Australasian gannets were determined using model averaging as the combined weight of the top set of models was low ($w_i > 0.9$)"

P4L41: How were classifications controlled for by the number of trips sampled for each individual?

Our methods state: "Individuals that spent >70% of trips during a deployment in PPB or BS were classified as PE-inshore and PE-pelagic, respectively, while individuals that displayed no preference were defined as PE-mixed." This means we classify each foraging trip to the corresponding habitat and then, the proportional number of trips performed in each habitat were estimated. Those deployments with more than 70% of the foraging trips made in a particular habitat got that same habitat classification. The number of trips sampled per individual was considered within the proportions estimations. The sampling was standardised among individuals by the number of days that the GPS and accelerometers were deployed on each bird (10-12 d).

Perhaps longer deployments resulted in more mixed classifications?

We are assuming this comment refers to the fact that longer deployments resulted in more trips, which we have addressed in previous comments. More trips obtained might result in obtaining more of a mix of pelagic and inshore trips, but because we only are taking into the account birds with at least three trips, and the number of trips in each habitat proportionally to the total number of trips obtained, we should still be avoiding this bias. . The deployment length for this study (10-12d) was

considered to be long enough to represent the foraging patterns of an individual at a certain time of the year.

The mixed classification also seems like a way to artificially inflate consistency in the foraging metrics. This should be strongly/more clearly justified or removed from the analysis.

What we are seeing here is a gradient from least consistent (pelagic individuals then birds with mixed strategy) to more consistent (inshore individuals). We are unsure why the reviewer suggested the mixed classification would artificially inflate consistency as mixed strategists would not be more consistent than inshore birds that tend to go to more similar places by, for example, following bathymetric features, as opposed to mixed strategists and pelagic birds, which will tend to visit less similar locations.

P5L4-6: A stronger 1st sentence of the discussion should be considered since the current one sheds no light on the interpretation, implications, or importance of the study.

The text has been modified and the discussion now starts with a reminder of why this study is important, before summarising the main results of the study.

P5L9-11: This seems like an oversimplification due to the omission of the mixed strategy birds.

We have now compared pelagic, vs mixed, vs inshore strategies to address this comment.

P5L12-14: This statement is repeated almost verbatim from the methods and points to an additional analysis that could be done with these data. It reads like a disclaimer. Move this somewhere lower in the discussion and provide context by suggesting what other environmental variables might be useful for this next step. Why would adding more environmental variables be useful? Given your results how would you approach this? Also, conclude this paragraph with a statement that summarizes the most important findings of this study.

This paragraph has now been deleted as the discussion has been rewritten to address concerns of clarity of the discussion.

P5L17-18: Given your topic sentence I thought you were going to justify the use of metrics in ecology. Rewrite to provide context for geographic variation.

The text has been modified.

P5L30: Omit the word “interestingly”. If you are discussing something it should be interesting. Rephrase the topic sentence of this paragraph to discuss links between foraging and breeding success.

The word “interestingly” has been deleted.

P5L39. Omit “In the present study” and rewrite topic sentence.

“In the present study” has now been deleted.

P5L39: How do your results show “temporal variation” in relation to the breeding stage? You are discussing differences between breeding stages, but “temporal variation” as mentioned here is very vague. Rephrase your topic sentence to discuss the importance of changes in foraging patterns during different reproductive stages.

The text has now been modified.

P5L49-57: Repeated paragraph.

The repeated paragraph has now been deleted.

P6L1-8: How does the repeatability of VeDBA factor into this discussion?

This part is not about the repeatability of VeDBA, but about WeDBA itself and relates to the sexual size dimorphism among gannets.

P6L10-14: Integrate this discussion topic with other results.

The discussion has now been rewritten.

Reviewer #2:

The manuscript titled “Influences on foraging behavior and consistency in a marine aerial predator” raises some interesting questions, but the study only goes part of the way to addressing them. Specifically, the Introduction raises interesting and significant ecological questions about the relationship between behavioral consistency, individual specialization, and the potential for adaptation in the context of climate change. The Introduction also alludes to the importance of understanding behavioral consistency as a basis for predictions about responses to environmental change. However, the study presented here is confined to a statistical analysis of the relationship between the foraging metrics of Australasian gannets (*Morus serrator*), fixed effects such as year,

colony, pelagic/nearshore habitat, sex, breeding stage, and random effects at the nest and individual levels.

The study does not incorporate any environmental factors that might explain variation in foraging metrics, and relies heavily on previous studies for interpretation. The main inference is that the apparent significance of individual random effects implies individual specialization and behavioral consistency. The main difference from similar studies of foraging metrics is that the same individuals were tagged on several occasions, allowing for comparisons between trips, breeding stage, and year that may provide information on the decay in behavioral consistency over time. The final conclusion that behavioral consistency could have important implications for population dynamics may well be true, but is based on the first two paragraphs of the Introduction rather than the study presented here. Similar questions about ecological traps due to behavioral consistency were raised and addressed by Sherley et al. *Curr. Biol.* 2017, but that paper is not cited here. Overall, the study falls short of the expectations raised in the Introduction. While it is useful to set out the broad ecological questions impression, it is also important to be clear about the specific question you will address in your study and what we will know at the end of the paper that was not known before.

Thank you for your comments. Parts of the paper have been rewritten to address your concerns (summary, introduction, discussion) and put less of an emphasis on the importance of understanding behavioural consistency to predict how species might adapt to environmental changes, with an emphasis on the link between habitat selection and behavioural consistency. By rewriting parts of the text, we made it clearer to the reader than one of the questions we wanted to investigate was whether behavioural consistency was driven by spatio-temporal factors or by intrinsic factors. Thank you for suggesting citing Sherley et al. 2017. This paper is now cited in the discussion.

Major comments:

My main methodological concern is with the removal of 'erroneous locations' based on a speed filter of 55 km/h. GPS is usually assumed to be sufficiently accurate that filtering of data points is rarely necessary. The implementation of the filter is unclear – the text suggests you removed any points that implied a speed of greater than 55 km/h, but if 55 km/h is an average speed, then speeds will sometimes be faster. Removal of datapoints will affect the distance traveled and the tortuosity index, so it becomes unclear whether differences in these metrics reflect genuine differences in foraging patterns or variation in GPS performance.

Erroneous locations are sometimes obtained from IgotU GPS data loggers and, therefore, a speed filter was used to remove this potential error. The data filter was applied to average speeds >55 km/h, which was selected based on Hamer et al. (2000, 2007) suggestion that this speed represented the upper limit in the average flying speed that a Northern gannet could reach and, in their research, this filter only accounted for 5% of GPS location records. In the absence of accurate speed data in Australasian gannets (a smaller species), 55 km/h was used as a filter. This resulted in the removal of only 3.4% of all the total GPS records.

It would be useful to see the delta-AICs between the selected fixed effects model and a mixed effects model with similar fixed structure for each foraging metric to confirm that the individual random effects are indeed supported by the data.

We followed the suggested protocols of Zuur et al. 2009 *Mixed Effects Models and Extensions in Ecology with R*, where the random structure should be tested with as many explanatory variables as possible in the fixed component, and tested whether the models with random effects were significantly different to those with fixed effects only using the *anova* function. Only those models where inclusion of the random factor explained more variance were then used in the second step of the variance components analysis.

Given that your question concerns the additional explanatory power of individual random effects, it's unclear whether model averaging is appropriate – why not simply use the fixed model with lowest AICc?

As indicated in Supplementary Table S1, the numerous models with $\Delta AIC < 4$ each had very little weight, making it difficult to justify selection of the model with the lowest AICc. We, therefore, took the conservative approach of exploring the fixed variables that had consistent influence through model averaging as recommended by Symonds ME, Moussalli A (2011) A brief guide to model selection, multimodel inference and model averaging in behavioural ecology using Akaike's information criterion. *Behav Ecol Sociobiol* 65:13-21.

In your second set of models, you are treating the output from the first set of models as data (P. 3; L 54). This is questionable without a clear method for propagating uncertainty.

This is incorrect, the second set of models does not use the output of the first set of models as data. Having identified the influence of individual variation on a parameter in the first models, the coefficients of variation (and standard deviations for bearings as these are circular measures) calculated from the raw data for each deployment were used as the response variables in the

second models to investigate the factors which might influence individual variation. The text has now been modified to make this point clearer.

What do you mean by “The dataset acquired was then modified ...” (P. 4; L 3)?

Apologies for the confusion. We meant to say ‘partitioned’. Only data from individuals that were tracked on consecutive trips, or during multiple breeding stages or multiple years were taken into account in this analysis. The text has now been modified to reflect this point.

You report a consistent decrease in variance associated with individual random effects over time (P. 4; LL 37-39), but how can we assess whether the reported decrease was significant? If this is your main result, it needs clearer support from the data and analysis.

Unfortunately, there is no way that we are aware of to know which values (proportion of total associated with the individual component) significantly differed between the timescales investigated. In this study, we used methods available in the literature to estimate trends over the different timescales studied. This is consistent with previous studies showing a decrease in foraging behaviour consistency over time (Harris et al., 2014, Camprasse et al. 2017). Despite the lack of methods to compare consistency values statistically, we still believe it is worth reporting on these differences between timescales as an observation only, similar to previous studies. References:

Harris S, Raya Rey A, Zavalaga C, Quintana F. 2014 Strong temporal consistency in the individual foraging behaviour of imperial shags *Phalacrocorax atriceps*. *Ibis*. 156, 523-533. (doi:10.1111/ibi.12159)

Camprasse ECM, Sutton GJ, Berlincourt M, Arnould JPY. 2017 Changing with the times: little penguins exhibit flexibility in foraging behaviour and low behavioural consistency. *Mar. Biol.* 164, 169. (doi:10.1007/s00227-017-3193-y)

Minor comments:

P. 1; L.34: Result here appears to differ from results presented on P 4. L 39.

We have corrected the range in the abstract.

P. 2; L 25 ff: The scope of work indicated here is far greater than achieved by this study.

Thank you for your comment. We have now modified the paper to reflect these concerns (abstract, introduction, discussion and conclusion) as mentioned above.

P 3; LL 6-7: Why?

We aimed for nest partners as this research is part of a wider study in which the comparison between pairs is important. To avoid confusion, we have now deleted this information from the methods. Although information is lacking on the similarity in foraging behaviour and consistency of breeding partners in vertebrates, research has shown that breeding partners can forage in more similar areas, and display more similar behaviour than expected by chance. Therefore, we wanted to account for any potential similarity between mates as both partners of breeding pairs were sampled most of the time. Nest identity as a random component did not appear to be significant, suggesting that breeding partners are not more similar in behaviour and consistency than expected by chance.

Reference:

Camprasse EC, Cherel Y, Arnould JP, Hoskins AJ, Bustamante P, Bost CA. 2017. Mate similarity in foraging Kerguelen shags: a combined bio-logging and stable isotope investigation. *Marine Ecology Progress Series* 31. 578:183-96.

P 5; L 49 ff: Duplicated paragraph.

The duplicated paragraph has now been deleted.

P 5; L 23: What do you mean by “habitat accessibility” here and elsewhere?

Thank you for your comment; “habitat accessibility” has now been deleted to avoid confusion.

P 6; L 6: How are you measuring dive rate and VeDBA here? If you are looking at number of dives per trip and average VeDBA over the course of a trip, then it would not be surprising that average VeDBA is lower on longer trips, despite more dives (see Boyd et al. *Ecol Model.* 2015). If, on the other hand, you are referring to total VeDBA per average trip, then yes it would be very surprising to find this is lower for larger-bodied animals on longer trips with more dives, and this would demand further examination.

Yes, we are measuring number of dives per foraging trip and average VeDBA per foraging trip as well. The word “surprisingly” has been removed.

P 6; L 13: I think you mean foraged in both habitats rather than displayed both behaviors. If not, how do you know whether they foraged alone or in a group?

Wells, et al. 2016 found (through animal-borne cameras) that gannets from Pope’s Eye foraging inside Port Phillip Bay were hunting alone. Subsequent unpublished camera data has verified this.

We, therefore, may assume that gannets from the present study are maintaining a similar

behaviour. However, to make this paragraph clearer we changed “displayed both behaviours” for “were able to forage in both habitats”.

Appendix B

Ref.: Ms. No. RSOS-181423.R2

Geographic, temporal and individual factors influencing foraging behaviour and consistency in Australasian gannets

Dear Reviewers and Editors,

We would like to use this opportunity to sincerely thank the reviewers for their detailed comments. They were all pertinent and useful for the improving of the manuscript. We have taken all these comments into consideration within the revised version. The corresponding changes and refinements made after revision are summarized in the following document 'Response to Referees'. Please find the editors and reviewers' comments in black and our answers are in blue.

Kind regards,

John Arnould (corresponding author)

Editor comments:

The authors have done a nice job of responding to reviewer comments in their revised submission. However, the two current reviewers raise additional questions that appear non-trivial. Some of the modifications suggested require re-analysis and/or revision of the conclusions. The data presented are valuable, but as one reviewer notes, without some clarification and more meaningful discussion the value of the results in terms of new information would be limited. The second reviewer also notes difficulty with accepting geometry of the habitat as a main factor; with a sample size of two, this could be a point well taken and I hope you will address this in your final revision.

I also suggest that you change your title to note "the Australian gannet" instead of "a marine aerial predator," which is vaguer. You might also specify the "factors" in the title so that readers know what to expect. Thanks and best wishes for your revision.

As suggested, we have modified the title to state "Australasian gannet". While we think it makes the title too wordy, we have also specified some of the "factors" in the title as requested.

Reviewer #3:

The manuscript is well written and tackles some very interesting ecological questions. While the argument that foraging behaviour will be more consistent in more stable environments is

convincing, it is not clear that the variability in behaviour resulting from increased environmental stability can be detected from tracking data. The data collection methods, authors' approach to quantifying and measuring the consistency of foraging behaviour are both thorough and appropriate.

My great concern is whether the results of the manuscript might be explained purely by the geometry of the habitat. The authors' analyses suggest that foraging behaviour is more consistent in benthic habitats than pelagic habitats, but to what degree are these results dependent on the geometry of the foraging region?

Consider the bearing metric and suppose the foraging region is a thin, elongated ellipse. If the colony is situated along the semi-major axis of the ellipse then the bearing will be highly constrained, but if the colony is situated on the semi-minor axis the bearing can be more variable. The authors need to discuss this issue before the manuscript is suitable for publication.

The issue of geometry is, indeed, valid and actually supports our findings. Individuals that had a narrow axis for foraging (PD animals) actually had less consistency than those feeding in the roughly circular Port Phillip Bay. Similarly, despite the arc of available headings to potential foraging areas being similar for the PE-pelagic and PE-inshore birds, the later were more consistent in their bearings from the colony. Text has been added in the Discussion to address this important issue.

In addition, there are some minor issues in relation to the choice and description of the methods.

As noted on P26L54, bearing is a circular variable. So standard modelling techniques will not be valid if the bearings encountered in practice span the 0/360 degree boundary. There should be some justification of the use of standard regression techniques for bearing.

Bearings have previously been used in standard modelling techniques, including ANOVAs and mixed models (Patrick and Weimerskirch, 2014; Waggitt et al., 2014; Potier et al., 2015; Sztukowski et al., 2018). The response variable does not have to be normal, but residuals of each model should be inspected to ensure they fitted the assumptions of the model (Potier et al., 2015; Zuur et al. 2009, p.20), which we used to allow us to determine the validity of our models.

References:

Patrick, S.C. and Weimerskirch, H., 2014. Consistency pays: sex differences and fitness consequences of behavioural specialization in a wide-ranging seabird. *Biology letters*, 10(10), p.20140630.

Potier, S., Carpentier, A., Grémillet, D., Leroy, B. and Lescroël, A., 2015. Individual repeatability of foraging behaviour in a marine predator, the great cormorant, *Phalacrocorax carbo*. *Animal Behaviour*, 103, pp.83-90.

Sztukowski, L.A., Cotton, P.A., Weimerskirch, H., Thompson, D.R., Torres, L.G., Sagar, P.M., Knights, A.M., Fayet, A.L. and Votier, S.C., 2018. Sex differences in individual foraging site fidelity of Campbell albatross. *Marine Ecology Progress Series*, 601, pp.227-238.

Waggitt, J.J., Briffa, M., Grecian, W.J., Newton, J., Patrick, S.C., Stauss, C. and Votier, S.C., 2014. Testing for sub-colony variation in seabird foraging behaviour: ecological and methodological consequences for understanding colonial living. *Marine Ecology Progress Series*, 498, pp.275-285.

Zuur, A., Ieno, E.N., Walker, N., Saveliev, A.A. and Smith, G.M., 2009. *Mixed effects models and extensions in ecology with R*. Springer Science & Business Media.

P26L1 It is the clustering that is unsupervised, not the wavelet transformation. I presume the intention was "The Ethnographer package was used to identify these behaviours by performing a k-means, unsupervised cluster analyses of one second windows of continuous wavelet spectra computed from the time series."

Yes, this is correct. Thank you for picking up our inadequate wording.

Reviewer: 4

Overall, the authors have done a nice work, revising the manuscript in accordance with the reviewer's comment. However, there is one critical concern regarding the interpretation of the results.

The authors state that the inshore foraging (PPB) exhibited more consistency, compared to the pelagic and mixed, in the foraging metrics, probably relating to its stability and predictability. However, it could be also because condition in foraging habitats in PPB is much less varied than these in Bass Strait (i.e. much smaller area to forage and narrower range of directions to travel in PPB: Fig. 1), which should restrict birds to occupy only specific habitats. The authors are probably true, in some parts, explaining that "seagrasses and sandy bottoms provide predictable stable habitats and features that can be used as cues for resource availability", but I also believe that comparing consistency in areas between LARGE and SMALL habitats is not fare to do (availability in habitat choice is very different). Please mention or explain this point in the Discussion. Otherwise, this study would be just to examine differences in foraging behavior of Australian gannets between sexes, breeding-stages, colonies, and years, which is not new and probably does not match to the description in the Introduction.

Whether comparing consistency in the pelagic and inshore habitats was simply comparing behaviour in small and large habitats is an important question. While the area used by PE-pelagic birds was 2-3 times that of PBB, the PE-inshore individuals represented <20% of the sampled population.

Consequently, PE-inshore birds are likely to have had greater areas of habitat per individual yet displayed greater behavioural consistency. Furthermore, the diversity of available habitats was greater within PBB than in Bass Strait yet individuals within the former were more consistent in their behaviour. Text has been added to the Discussion to cover this issue.

Minor comment:

"the average Vectorial Dynamic Body Acceleration (VeDBA) per activity and then all activity values were averaged within each foraging trip" (P4L4-6 in the cleared version of the revised ms).

> Why did you use the “averaged” values rather than the “sum” of VeDBA?

The sum of VeDBA (I presume you are referring here to the integral of the VeDBA time series) is analogous to total energy expenditure, whereas mean VeDBA is analogous to metabolic rate. We were interested in the latter (especially as the former would be correlated to time i.e. sampling duration). See:

Ladds, M.A., Rosen, D.A.S., Slip, D.J. et al. Proxies of energy expenditure for marine mammals: an experimental test of “the time trap”. *Sci Rep* 7, 11815 (2017). <https://doi.org/10.1038/s41598-017-11576-4>

> Do you mean “per activity” as “per diving”?

Thank you for picking up on this inadequate description of the methods. We have modified the text to clarify what was done.